# A team of heterochromatin factors collaborates with small RNA pathways to combat repetitive elements and germline stress

Alicia N McMurchy[†], Przemyslaw Stempor[†], Tessa Gaarenstroom[†], Brian Wysolmerski, Yan Dong, Darya Aussianikava, Alex Appert, Ni Huang, Paulina Kolasinska-Zwierz, Alexandra Sapetschnig, Eric A Miska, Julie Ahringer*

The Gurdon Institute and Department of Genetics, University of Cambridge, Cambridge, United Kingdom

**Abstract** Repetitive sequences derived from transposons make up a large fraction of eukaryotic genomes and must be silenced to protect genome integrity. Repetitive elements are often found in heterochromatin; however, the roles and interactions of heterochromatin proteins in repeat regulation are poorly understood. Here we show that a diverse set of *C. elegans* heterochromatin proteins act together with the piRNA and nuclear RNAi pathways to silence repetitive elements and prevent genotoxic stress in the germ line. Mutants in genes encoding HPL-2/HP1, LIN-13, LIN-61, LET-418/Mi-2, and H3K9me2 histone methyltransferase MET-2/SETDB1 also show functionally redundant sterility, increased germline apoptosis, DNA repair defects, and interactions with small RNA pathways. Remarkably, fertility of heterochromatin mutants could be partially restored by inhibiting *cep-1*/p53, endogenous meiotic double strand breaks, or the expression of MIRAGE1 DNA transposons. Functional redundancy among factors and pathways underlies the importance of safeguarding the genome through multiple means.

*For correspondence: ja219@cam.ac.uk

[†]These authors contributed equally to this work

## Introduction

Heterochromatin, the more tightly packed form of chromatin, plays important roles in maintaining the structural and functional integrity of the genome (*Wang et al., 2016*). It is less transcriptionally active than euchromatin and highly enriched for repetitive elements such as transposons and satellite repeats, which are kept silent to maintain genome integrity. The heterochromatin state is stable and heritable across generations highlighting the importance of keeping certain regions of the genome repressed.

Histones in heterochromatin are marked with modifications associated with transcriptional repression such as H3K9 methylation. In organisms with point centromeres, heterochromatin is typically found in large domains adjacent to centromeres and telomeres (*Wang et al., 2016*). In *C. elegans*, heterochromatin associated histone methylations H3K9me2 and H3K9me3 are instead mostly found in many small domains on the distal arm regions of autosomal chromosomes (*Liu et al., 2011*). This pattern is likely to be related to the holocentric nature of *C. elegans* chromosomes, which have distributed centromeres rather than a single point centromere. Two histone methyltransferases carry out all H3K9 methylation (*Towbin et al., 2012*). The SETDB1 homolog MET-2 carries out mono- and di-methylation of H3K9. SET-25 primarily carries out tri-methylation of H3K9, but it can generate all three methylated forms of H3K9. In the absence of both proteins, H3K9 methylation is undetectable,

heterochromatic distal arm regions show reduced association with the nuclear lamina, and hetero-chromatic transgenes are desilenced (*Towbin et al., 2012*).

A hallmark of heterochromatin is heterochromatin protein 1 (HP1), the first heterochromatin protein to be discovered through work in Drosophila (*Zeng et al., 2010*; *James and Elgin, 1986*). HP1 contains a chromodomain that binds to methylated H3K9, and it is essential for heterochromatin maintenance (*Zeng et al., 2010*). In addition to HP1, a large and diverse array of proteins is associated with heterochromatin, including nucleosome remodelers, histone modifying enzymes, histone binding proteins, and DNA binding proteins (*Saksouk et al., 2015*; *Meier and Brehm, 2014*). However, the functions and interactions of heterochromatin proteins are not well understood.

Many *C. elegans* proteins that have predicted functions in heterochromatin or transcriptional repression are important for development. These include MET-2/SETDB1, HPL-2/HP1, LIN-61, LIN-13, and LET-418/Mi-2 (8–13). HPL-2 is a *C. elegans* ortholog of heterochromatin protein HP1, and LIN-61 is a protein containing MBT (malignant brain tumor) repeats. Both HPL-2 and LIN-61 can bind to all methylated forms of H3K9 in vitro (*Koester-Eiserfunke and Fischle, 2011*; *Garrigues et al., 2015*; *Studencka et al., 2012*), and both can repress a heterochromatic reporter (*Towbin et al., 2012*; *Couteau et al., 2002*; *Harrison et al., 2007*). LIN-13 is a multi-zinc finger protein (*Meléndez and Greenwald, 2000*). A complex containing LIN-13, HPL-2, and LIN-61 has been detected in vivo, and LIN-13 is required for the formation of HPL-2::GFP nuclear foci (*Wu et al., 2012*; *Coustham et al., 2006*). LET-418 is an ortholog of Mi-2, an ATP-dependent nucleosome remodelling component of the repressive NuRD and Mec complexes (*von Zelewsky et al., 2000*; *Unhavaithaya et al., 2002*; *Passannante et al., 2010*).

Mutants of *hpl-2*, *lin-61*, *lin-13*, *met-2*, and *let-418* display both germ line and somatic defects. *let-418* and *lin-13* null mutants are sterile (*von Zelewsky et al., 2000*; *Meléndez and Greenwald, 2000*), *hpl-2* null mutants show temperature sensitive sterility (*Schott et al., 2006*), and *lin-61* and *met-2* null mutants have slightly reduced brood sizes (*Koester-Eiserfunke and Fischle, 2011*). The underlying cause of the fertility defects is not known, but *hpl-2* mutants have been shown to produce abnormal oocytes, suggesting defective gametogenesis (*Couteau et al., 2002*). Somatic defects are pleiotropic and show similarities among mutants, with most showing slow growth, somatic expression of germ line genes, synthetic vulval development defects, and larval arrest (some only at high temperature) (*Meléndez and Greenwald, 2000*; *Schott et al., 2006*; *Wu et al., 2012*; *Harrison et al., 2007*; *Coustham et al., 2006*; *Unhavaithaya et al., 2002*; *Andersen and Horvitz, 2007*; *Kerr et al., 2014*; *Petrella et al., 2011*; *Poulin et al., 2005*). Additionally, genetic interactions have been observed between some of the mutants, suggesting partially redundant functions, and that defects may result from alteration of a shared heterochromatin-linked process (*Koester-Eiserfunke and Fischle, 2011*; *Coustham et al., 2006*; *Simonet et al., 2007*).

The genomic distribution of only one of the above heterochromatin proteins has been studied. An HPL-2 ChIP-chip study in early embryos showed that most binding was on the distal arm regions of autosomes in a pattern of similar to H3K9me1 and H3K9me2; interestingly, binding to chromatin was not dependent on H3K9 methylation (*Garrigues et al., 2015*). HPL-2 was observed to be broadly genic, with additional association at promoters in central chromosome regions and at repeats on distal arm regions; however, no clear relationship between HPL-2 binding and gene expression changes was observed (*Garrigues et al., 2015*). Systematic and comparative analyses of heterochromatin factors are needed to understand their functions.

The genomic binding patterns of orthologs of some of the above factors suggest roles in the regulation of mobile elements. SETDB1 binds to promoters of developmentally regulated genes in mammalian embryonic stem cells, 40% of which are found next to or overlapping endogenous retroviruses (*Yuan et al., 2009*; *Karimi et al., 2011*; *Bilodeau et al., 2009*). In addition, retrotransposons are derepressed in *Setdb1* knockout mouse ES cell lines and primordial germ cells (*Karimi et al., 2011*; *Liu et al., 2014*). Retrotransposons are also repressed by HP1α and HP1β in mESCs, but it is a different set of retrotransposons than is targeted by SETDB1 (*Maksakova et al., 2013*). Drosophila HP1 binds to genes and to transposable elements, particularly in pericentric chromosomal regions, but transposable element expression in mutants has not been assessed (*Greil et al., 2003*; *de Wit et al., 2005*). A recent study showed that Mi-2 could bind to a LINE1 retrotransposon promoter and repress a LINE1 reporter in human and mouse cell lines (*Montoya-Durango et al., 2016*). The genomic distribution of Mi-2 is unclear as different genome-wide binding studies in human ES cells have yielded conflicting results (*Hu and Wade, 2012*).

The expression of repetitive elements can be detrimental to genome stability due to the negative effects of homologous recombination and transposon-induced breaks. Because the germline produces the gametes that transmit genetic information across generations, silencing of repetitive elements is an absolute requirement for germ line health. A small RNA pathway called the piRNA pathway, present in most animals, plays a role in transposon silencing in the germ line (*Weick and Miska, 2014*). Recent work in *C. elegans* implicated HPL-2/HP1 and the H3K9me3 histone methyltransferase SET-25 in piRNA pathway function, indicating a connection between heterochromatin and small RNA silencing of piRNA targets (*Ashe et al., 2012*).

In the *C. elegans* germ line, the piRNA pathway involves generation of 21nt piRNAs that bind to the Piwi argonaute protein PRG-1 (*Wang and Reinke, 2008*; *Das et al., 2008*; *Batista et al., 2008*). This triggers generation of secondary 22G siRNAs that mediate silencing either in the cytoplasm or nucleus (*Ashe et al., 2012*; *Das et al., 2008*; *Shirayama et al., 2012*; *Luteijn et al., 2012*; *Lee et al., 2012*; *Guang et al., 2010*; *Gu et al., 2009, 2012*; *Burton et al., 2011*; *Burkhart et al., 2011*; *Buckley et al., 2012*). Cytoplasmic silencing mechanisms are not well understood, but recent advances have been made in the understanding of transcriptional silencing. In the nucleus, the piRNA pathway engages a second small RNA pathway called the nuclear RNAi pathway (or nrde pathway), which orchestrates H3K9 and H3K27 methylation and/or inhibition of RNA Pol II (*Guang et al., 2010*; *Burkhart et al., 2011*; *Mao et al., 2015*; *Alló and Kornblihtt, 2010*). Although the nrde pathway can be triggered by the piRNA pathway in the germ line, it is also active in the soma, with dedicated Argonaute proteins for germ line (HRDE-1) and soma (NRDE-3) (*Guang et al., 2010*; *Burkhart et al., 2011*; *Buckley et al., 2012*; *Guang et al., 2008*). Regions transcriptionally upregulated in *hrde-1* mutants were found to be enriched for retrotransposons, suggesting that repetitive elements may be endogenous targets in the germ line (*Ni et al., 2014*).

In addition to silencing transcription and maintaining the structural and functional integrity of the genome, heterochromatin also plays an important role in DNA repair. Heterochromatic compaction protects DNA from damage, and regulated decondensation is important for damage repair (*Feng et al., 2016*). Additionally, in mammals, transient formation of heterochromatin occurs at the edges of double strand breaks, which involves recruitment of heterochromatin-associated proteins HP1 and nucleosome remodeller Mi-2, as well as methylation of H3K9 (*Gursoy-Yuzugullu et al., 2016*). This is thought to aid in damage repair by keeping the broken strands in proximity and inhibiting local transcription.

Here, through systematic genetic and genomic analyses, we investigate interactions and functions of five heterochromatin proteins (HPL-2/HP1, LIN-13, LIN-61, LET-418/Mi-2 and MET-2/SETDB1) and relationships with the piRNA and nuclear RNAi pathways. Our results reveal a nexus of factors that cooperate to prevent expression of repetitive elements and protect the germ line from endogenous damage.

## Results

### Heterochromatin factors show partially redundant functions for fertility

The five genes we study here (*hpl-2*/HP1, *lin-13*, *lin-61*, *let-418*/Mi-2, and *met-2*) are needed for normal fertility ([*von Zelewsky et al., 2000*; *Meléndez and Greenwald, 2000*; *Schott et al., 2006*; *Koester-Eiserfunke and Fischle, 2011*; *Ceol et al., 2006*; *Thomas et al., 2003*]; *Supplementary file 1*). Previous analyses uncovered genetic interactions in fertility between three pairs of genes (*hpl-2* and *lin-13*, *lin-61* and *hpl-2*, *lin-61* and *met-2*; (*Koester-Eiserfunke and Fischle, 2011*; *Coustham et al., 2006*; *Simonet et al., 2007*); however, the remaining six combinations were untested. Using RNAi in mutant backgrounds, we found that double loss of function of each of the uninvestigated pairs also caused synthetic sterility, which we also observed for three tested double mutant combinations (*Figure 1A–C*). The single mutants show complex and pleiotropic germ line defects, but they all showed a high occurrence of abnormal oocytes, suggesting that sterility may be due in part to abnormal oogenesis (*Figure 1—figure supplement 1*). The fertility defects of single mutants and the enhancement in double loss of function combinations indicate that the five heterochromatin factors each have unique and partially redundant germ line roles. The genetic interactions suggest that the five factors may cooperate in a germ line process required for fertility.

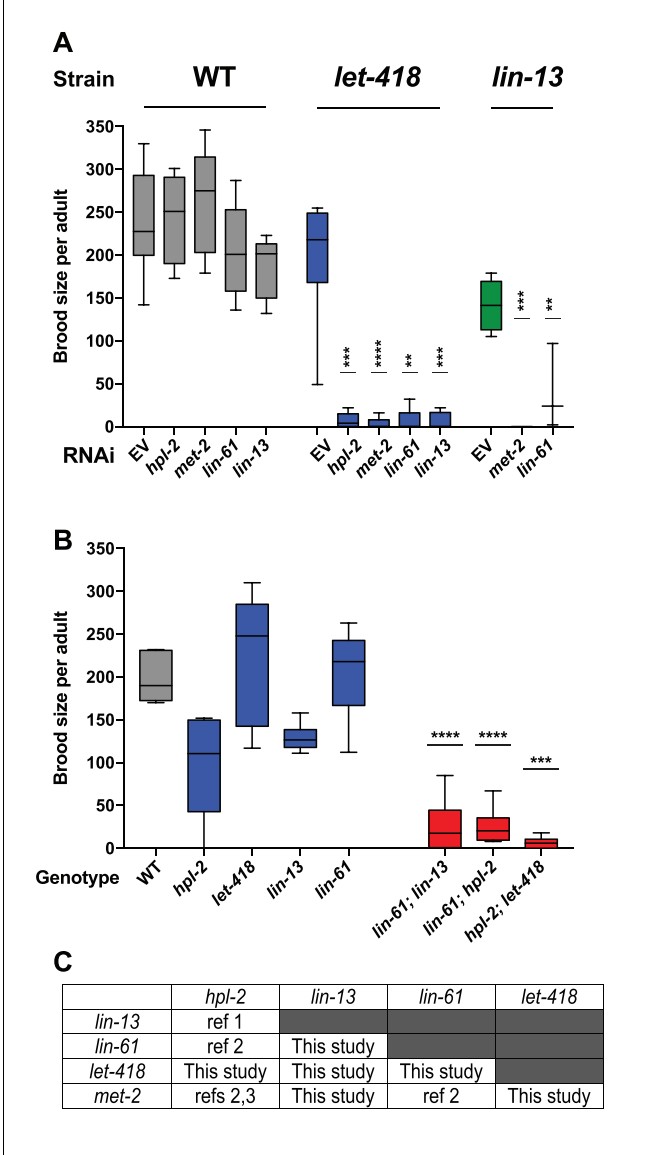

**Figure 1.** Heterochromatin proteins have redundant roles in fertility. (**A**) Genetic interactions in fertility assayed using RNAi. Indicated RNAi of wild-type, *let-418(n3536)*, *or lin-13(n770)* was conducted by feeding at 20°C as described in the methods. Results are a combination of two independent experiments with the progeny of 3–8 total broods counted for each strain/RNAi combination. A one-sided t-test was used to determine whether the mutant/RNAi combination had a lower brood size than expected under a multiplicative model of interaction when compared to the mutant grown on empty vector RNAi and the individual RNAi knockdowns in wild-type animals. Brood size is significantly lower than expected for all RNAi/mutant combinations at p<0.05. (**B**) Indicated double mutants were constructed and their brood sizes compared to that of the individual signal mutants raised at 20°C. Statistical testing was as in (**A**), with brood sizes of the three double mutants significantly lower than expected at p<0.05 in a one-sided t-test. (**C**) All pairs of *hpl-2, lin-13, lin-61, let-418*, and *met-2* show genetic interactions in fertility, as determined in this study or previous studies. ref 1. Coustam et al, *Dev Biol.* 2006. ref 2. Koester-Eiserfunke and Fischle, *PLoS Genet.* 2011. ref 3. Simonet et al, *Dev Biol.* 2007. **Supplementary file 1** shows previously reported sterility phenotypes. **Figure 1—figure supplement 1** shows examples and quantification of abnormal oogenesis in heterochromatin mutants.

The following figure supplement is available for figure 1:

**Figure supplement 1.** Heterochromatin mutants display abnormal oogenesis.

## HPL-2, LIN-13, LIN-61, LET-418, and MET-2 are enriched at repetitive elements and show extensive co-binding

To begin to investigate the roles of HPL-2, LIN-13, LIN-61, LET-418, and MET-2 in genome regulation, we mapped their binding locations using ChIP-seq analyses in young adults and compared the patterns to each other and to those of H3K9me2 and H3K9me3. Binding of each of the five factors is enriched on the distal arm regions of the autosomes (*Figure 2A*; *Figure 2—figure supplement 1A*), as previously seen for H3K9me2, H3K9me3, and HPL-2 (*Garrigues et al., 2015*; *Liu et al., 2011*; *Sha et al., 2010*). Examination of ChIP-seq signals at a more local level revealed similar binding patterns for the five heterochromatin proteins (*Figure 2B*). Indeed, genome-wide correlation analyses showed significant positive correlations in signal between all datasets (*Figure 2—figure supplement 1B*). In addition, signals for each of the five heterochromatin factors showed high correlation with H3K9me2 but not with H3K9me3 (*Figure 2B*, *Figure 2—figure supplement 1B*).

To further investigate patterns of binding, we identified regions of peak enrichment for each dataset (*Figure 2—figure supplement 1C*, *Figure 2—source data 1*; 12449 to 19313 peaks per factor). For each factor, peaks are enriched on the distal chromosomal regions of autosomes; most peaks are intergenic or located in introns, with enrichment for intergenic binding in central chromosomal regions and enrichment for intronic binding in distal arm regions (*Figure 2—figure supplement 1A*).

To facilitate comparisons between datasets, we merged peak calls from all factors into a superset termed Any5 (n = 33,301), then annotated each region in the Any5 set for the factors bound (*Figure 2—source data 1*). There is a high degree of peak overlap among the five factors, with 58% of sites in the Any5 set being bound by >1 factor (*Figure 2C*). Sites uniquely bound by only one factor are in the minority within each dataset (3.4–27.4%, *Figure 2C* and *Figure 2—figure supplement 1C*). Strikingly, the largest binding group contains all five factors (termed 'All5'; *Figure 2C*). Enrichments for H3K9me2 and H3K9me3 vary between binding classes, with the All5 class showing high enrichment for H3K9me2 (*Figure 2C*). These results show that HPL-2, LET-418, LIN-13, LIN-61, and MET-2 extensively overlap in binding genome-wide.

The previous HPL-2 ChIP-chip study in embryos noted binding at repetitive elements, which are concentrated on the distal arm regions of autosomes (*Garrigues et al., 2015*). Repeat-rich heterochromatin in *C. elegans* is distributed in small domains rather than being concentrated in large regions as in mammals or Drosophila; therefore, the sequences of most repetitive regions have been determined. To investigate the association of heterochromatin factor binding at repetitive DNA, we used the recent Dfam2.0 annotation, which classified 62,331 individual repetitive elements in *C. elegans* into 184 repeat families, which were further classified by type (e.g., DNA transposon, retrotransposon, satellite, or unknown [*Hubley et al., 2016*]).

We observed that HPL-2, LIN-13, LIN-61, LET-418, and MET-2 are all strongly associated with repetitive DNA elements (*Figure 2C*). A large proportion of each factor's peaks overlaps a repeat sequence (46.3–71.0%), and regions with all five factors have particularly strong repeat association (76.6%, *Figure 2C*, *Figure 2—figure supplement 1C*). Furthermore, of the total set of 62331 annotated repetitive elements, nearly half (46%) overlap a peak of at least one factor, and 8002 (13%) overlap all five factors (*Figure 2—figure supplement 1D*). All repeat types and 180 of 184 repeat families are associated with a heterochromatin factor peak; of these, 105 repeat families are enriched for binding by at least one factor (*Figure 2—source data 2*). HPL-2, LIN-13, LIN-61, MET-2, and H3K9me2 have a particularly strong association with Helitron families; LET-418 shows generally lower enrichment on repeat families than the other factors (*Figure 2D*; *Figure 2—figure supplements 2–6*).

H3K9me2 and H3K9me3 show different patterns of repeat enrichment. H3K9me2 is more associated with DNA transposons and satellite repeats, similar to the heterochromatin factors, whereas H3K9me3 is particularly associated with retrotransposon families, especially LINE and SINE elements (*Figure 2D*; *Figure 2—figure supplements 7–8*). We also observed that H3K9me2 and all heterochromatin factors are enriched at telomeres, whereas H3K9me3 is not (*Figure 2—figure supplement 1E*). The binding and co-association of HPL-2, LET-418, LIN-13, LIN-61, and MET-2 at repetitive elements suggests roles in the regulation of these sequences.

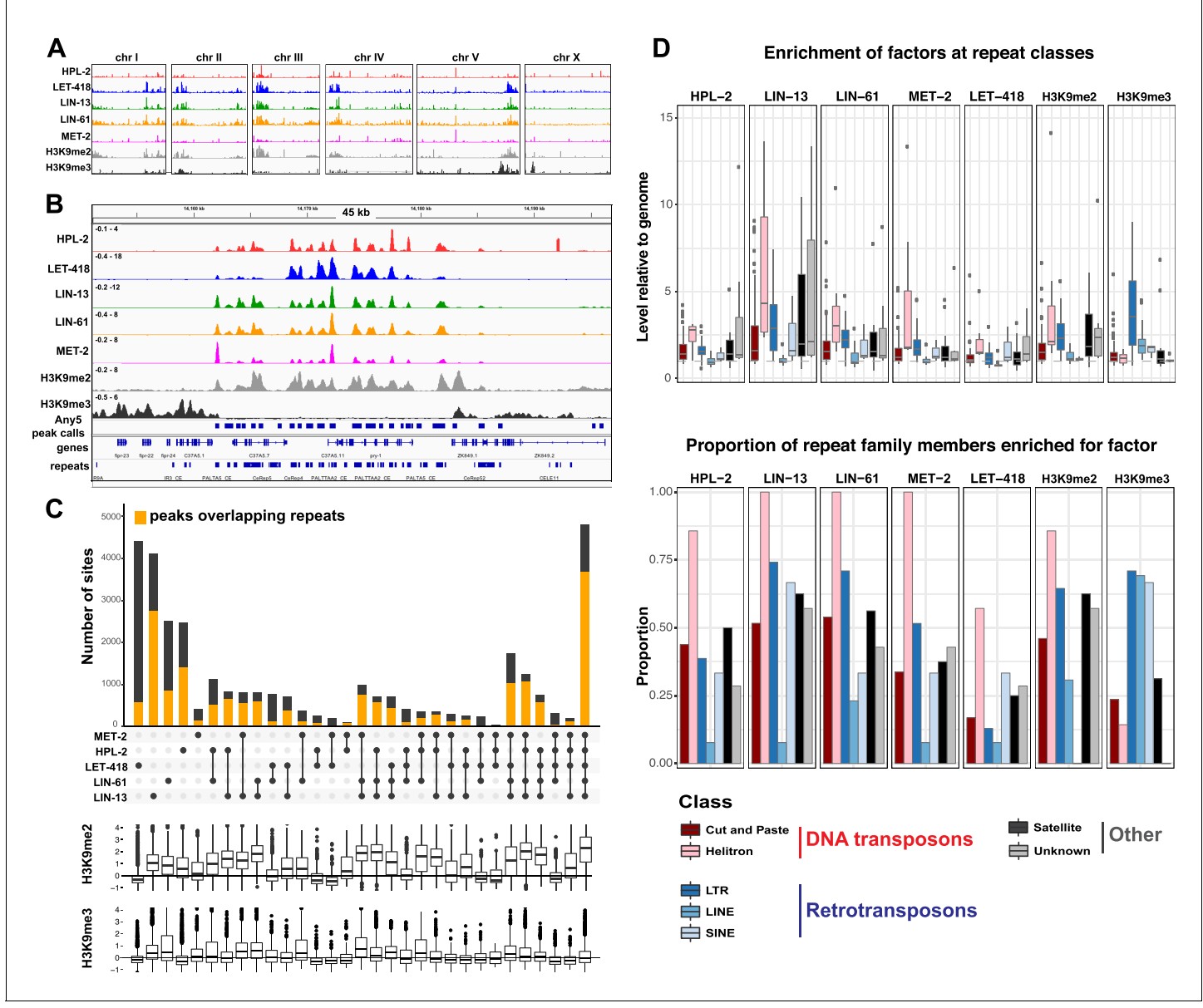

**Figure 2.** HPL-2, LET-418, LIN-13, LIN-61, and MET-2 show extensive co-binding and are enriched at repetitive elements. (**A**) Distribution of the indicated proteins and histone modifications over each *C. elegans* chromosome. z-scored ChIP-seq tracks are shown for HPL-2 (red), LET-418 (blue), LIN-13 (green), LIN-61 (orange), MET-2 (pink), H3K9me2 (grey) and H3K9me3 (black) on each chromosome, demonstrating enrichment over the autosomal arms. *Figure 2—figure supplement 1A* shows distributions of peak locations in different chromosome regions. (**B**) IGV browser screenshot showing similar patterns of the heterochromatin factors and H3K9me2 methylation over a 45 kb region containing multiple repeat elements. z-scored ChIP-seq tracks are as in (**A**). Any5 peak calls denote combined peak calls for any of the five proteins; repeats are from Dfam2.0 (*Hubley et al., 2016*). *Figure 2—figure supplement 1B* shows correlations in signal between all datasets. (**C**) UpSet plot of the association of heterochromatin factors with the 33,301 Any5 peak calls. Dots indicate peak class is bound by the factor. Bars show total number of peaks per class, the orange portion denoting overlap with repeat elements. Below the bar chart relative enrichments for H3K9me2 and H3K9me3 are shown. The peaks that overlap all five factors constitute the largest class (n = 4810). *Figure 2—figure supplement 1* gives total peak numbers per factor, number of peaks overlapping repeats, and number of repeats bound by each factor. *Figure 2—source data 1* gives peak calls. (**D**) Associations of factors and repeat classes. Upper panel: levels of indicated protein or histone modification on families within indicated Dfam 2.0 repeat classes relative to the genome average. Bottom panel: Proportion of families within each repeat class significantly enriched for indicated factor or histone modification. Criteria for enrichment are >1.5 fold mean enrichment of family relative to genome, FDR < 0.1, considering families with at least 10 members. Number of families with 10 or more members within each class are: Cut and paste (n = 89), Helitron (n = 7), LTR (n = 31), LINE (n = 13), SINE (n = 3), Satellite (n = 16), Unknown (n = 7). *Figure 2—source data 2* gives enrichment scores for repeat family factor binding.

*Figure 2 continued on next page*

*Figure 2 continued*

The following source data and figure supplements are available for figure 2:

**Source data 1.** Peak calls.
**Source data 2.** Enrichment of factors at repeat families.
**Source data 3.** Alignment Statistics for ChIP and RNA sequencing.
**Figure supplement 1.** Correlation of HPL-2, LET-418, LIN-13, LIN-61, and MET-2 ChIP-seq tracks and enrichment on chromosome arms, repetitive elements, and telomeres.
**Figure supplement 2.** Enrichment of HPL-2 at repeat families.
**Figure supplement 3.** Enrichment of LIN-61 at repeat families.
**Figure supplement 4.** Enrichment of MET-2 at repeat families.
**Figure supplement 5.** Enrichment of LIN-13 at repeat families.
**Figure supplement 6.** Enrichment of LET-418 at repeat families.
**Figure supplement 7.** Enrichment of H3K9me2 at repeat families.
**Figure supplement 8.** Enrichment of H3K9me3 at repeat families.

## Repetitive elements are desilenced in *hpl-2*, *lin-13*, *lin-61*, *let-418*, and *met-2 set-25* mutants

Because silencing of repetitive DNA elements is important for germ line function, we considered that the heterochromatin factors might function in preventing repeat expression. To investigate this possibility, we generated and analysed RNA sequence expression data for wild-type and heterochromatin mutant adults. MET-2 deposits H3K9me1 and H3K9me2, but all three methylation states of H3K9 are still present in *met-2* mutants (at lower levels) due to the action of SET-25 (*Towbin et al., 2012*). Therefore we assayed a *met-2 set-25* double mutant, in which H3K9 methylation was not detectable (*Towbin et al., 2012*). For each strain, we performed two biological replicates and differential expression analyses of the 62,331 Dfam2.0 repeat elements.

We observed upregulation of repetitive elements in every mutant strain (*hpl-2*, *let-418*, *lin-13*, *lin-61*, and *met-2 set-25*) (*Figure 3A–D*, *Figure 3—figure supplement 1*, *Figure 3—source data 1*). A total of 71 individual repeat elements representing 29 different families were upregulated in at least one mutant (*Figure 3—source data 2*; upregulation of 61/71 individual elements was confirmed based on uniquely mapping reads, see Materials and methods). We observed a striking overlap in the sets of repetitive elements regulated by the heterochromatin factors: 41% of elements are upregulated in more than one mutant (*Figure 3A*; *Figure 3—source data 2*). Furthermore, seven repeat elements are upregulated in all five strains, all of which are MIRAGE1 DNA transposable elements (*Figure 3A,C,D*, *Figure 3—source data 2*). The majority of repetitive elements upregulated in each heterochromatin mutant strain are DNA transposons, but retrotransposons are enriched for being upregulated (*Figure 3B*). Mutants show variation in the classes of repeats regulated; for example, SINE retrotransposons are particularly affected in *let-418* mutants, while many Helitron elements are upregulated in *lin-13* mutants (*Figure 3B*).

Overall, the total number of individual repetitive elements found with significantly altered expression is extremely small (<1%) relative to the >30,000 with factor binding, indicating that binding does not generally regulate repeat transcription. There are many types of repetitive elements, and only a small fraction would be expected to have potential for RNA expression. For example, 67% of repetitive elements are predicted to be non-autonomous DNA transposons, which would be mobilised in trans by a transposase encoded by another repetitive element, and many annotated

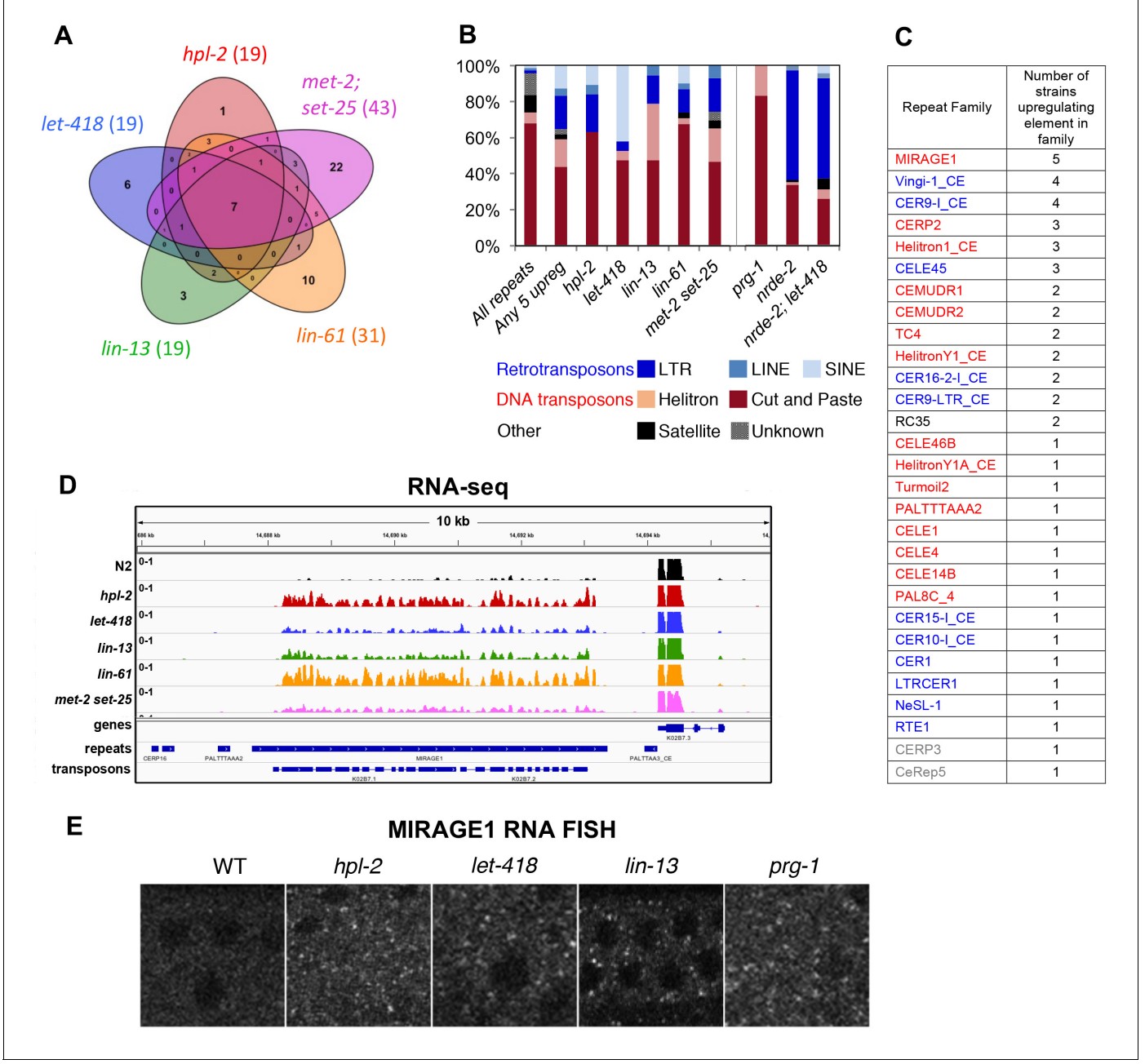

**Figure 3.** Repetitive elements are upregulated in *hpl-2, let-418, lin-13, lin-61, met-2 set-25, prg-1, nrde-2,* and *nrde-2; let-418* mutants. (A) Venn diagram of elements upregulated in *hpl-2, let-418, lin-13, lin-61,* and *met-2 set-25* mutants. The seven elements upregulated in all five strains are MIRAGE1 elements. (B) Distribution of Dfam 2.0 repeat classes upregulated in each strain. (C) Repeat families with members upregulated in at least one of *hpl-2, let-418, lin-13, lin-61,* or *met-2 set-25* mutant strains. (D) IGV browser screenshot of a MIRAGE1 element that is upregulated in all mutant strains. Tracks are reads per million of two combined replicates. *Figure 3—figure supplement 1* gives further examples of elements upregulated in different strains. (E) Single molecule RNA-FISH signals of MIRAGE one element RNA in the adult germ line (white dots). Signal is not detectable in wild-type but is abundant in the indicated mutant backgrounds. *Figure 3—figure supplement 3* shows additional images of MIRAGE1 and *sqv-1* control RNA FISH in germline and somatic tissues. *Figure 3—figure supplement 2* shows enrichment of heterochromatin factors, H3K9me2, and H3K9me3 on regulated genes and repeats.

The following source data and figure supplements are available for figure 3:

**Source data 1.** Analysis of repeats.

**Source data 2.** Repeats upregulated in any mutant strain.

*Figure 3 continued on next page*

*Figure 3 continued*

**Source data 3.** Analysis of genes.

**Figure supplement 1.** Examples of unique and co-regulated repeat elements in various heterochromatin mutants.

**Figure supplement 2.** Heterochromatin factors and H3K9 methylation show enriched association with upregulated genes and repeats.

**Figure supplement 3.** MIRAGE1 RNA is upregulated in the germ lines of *hpl-2, lin-13, let-418,* and *prg-1* mutants.

elements are small fragments of larger elements. We therefore wondered whether the upregulation of repetitive element expression in heterochromatin mutants might be particularly associated with transposases or retrotransposons. Of 62331 Dfam2.0 elements, 221 overlap a predicted transposase ORF, and 1085 are annotated as LTR retrotransposons. We found that elements upregulated in any heterochromatin factor mutant are 83-fold enriched for containing a transposase (21 of 71) and 10-fold enriched for LTR retrotransposons (13 of 71), together accounting for nearly half of upregulated repeats. Therefore, a key role of heterochromatin factors is to suppress expression of repetitive element transposases. The widespread binding of heterochromatin factors to non-expressed repetitive elements is likely to play roles other than in the regulation of transcription. These could include preventing the cutting, copying, or movement of elements, or maintaining genome integrity by supressing homologous recombination between repetitive elements (*Chiolo et al., 2011*; *Sinha et al., 2009*).

We also analysed alterations in protein coding gene expression in the heterochromatin mutants. Consistent with roles in repression, we identified three to five times more genes with upregulated expression in each mutant strain compared to those with reduced expression (267–513 upregulated genes per mutant; *Figure 3—source data 3*). Additionally, there is a high degree of overlap among the upregulated genes; of the total set of 1155 genes upregulated in any of the five mutant strains, 404 are upregulated in more than one (*Figure 3—source data 3*). Heterochromatin factors are enriched at upregulated genes, but not downregulated genes (except for genes misregulated in *let-418* mutants); additionally, both H3K9me2 and H3K9me3 are enriched at upregulated genes in all mutant strains (*Figure 3—figure supplement 2*). Enrichment for all factors and H3K9 methylation is particularly strong at genes upregulated in *met-2 set-25* mutants (*Figure 3—figure supplement 2*). These associations suggest direct roles in repression.

## Repression of desilenced MIRAGE1 elements partially restores fertility of heterochromatin mutants

The silencing of repetitive elements is a universal conserved feature of germ line function. The prominent upregulation of MIRAGE1 elements in all heterochromatin mutants prompted us to ask whether expression of this element might play a role in their reduced fertility. MIRAGE1 is an autonomous DNA transposable element that has two open reading frames. Of 69 MIRAGE1 element annotations in Dfam2.0, only six are full length. These six, plus an additional six partial MIRAGE1 elements are upregulated in at least one heterochromatin mutant, and both ORFs show upregulation.

We first examined the tissue distribution of MIRAGE RNA using RNA-FISH (*Raj et al., 2008*). As expected from the RNA-seq results, wild-type adults had very low levels of MIRAGE1 RNA-FISH signal in germ line and soma (*Figure 3E* and *Figure 3—figure supplement 3*). In three tested heterochromatin mutants (*hpl-2, let-418,* and *lin-13*), we observed abundant germ line localized MIRAGE1 RNA whereas somatic expression remained low (*Figure 3E* and *Figure 3—figure supplement 3*). Therefore, *hpl-2, let-418,* and *lin-13* are important for repression of MIRAGE1 in the germ line.

To test whether upregulation of MIRAGE1 contributes to sterility, we used two sets of RNAi clones to simultaneously knockdown ORF1 and ORF2 (sets termed *mirage-A* and *mirage-B*). *mirage-A* and *mirage-B* target 16 different MIRAGE1 elements, including all full length elements and most of the MIRAGE1 elements upregulated in each of these mutants (8/8 for *let-418*, 7/8 for *lin-13*, and 10/11 for *hpl-2*). To assess an effect on fertility, we grew *hpl-2, lin-13,* and *let-418* mutants at 25°C, a condition under which they are nearly sterile, and tested for an increase in brood size after RNAi

knockdown. Remarkably, RNAi of MIRAGE1 using *mirage-A* or *mirage-B* sets of RNAi clones led to a small but significant increase in fertility of all three mutants, showing that inappropriate MIRAGE1 expression contributes to their sterility (*Figure 4A*). We also observed that MIRAGE1 RNAi resulted in amelioration of somatic growth defects (not shown). These results indicate that one mechanism by

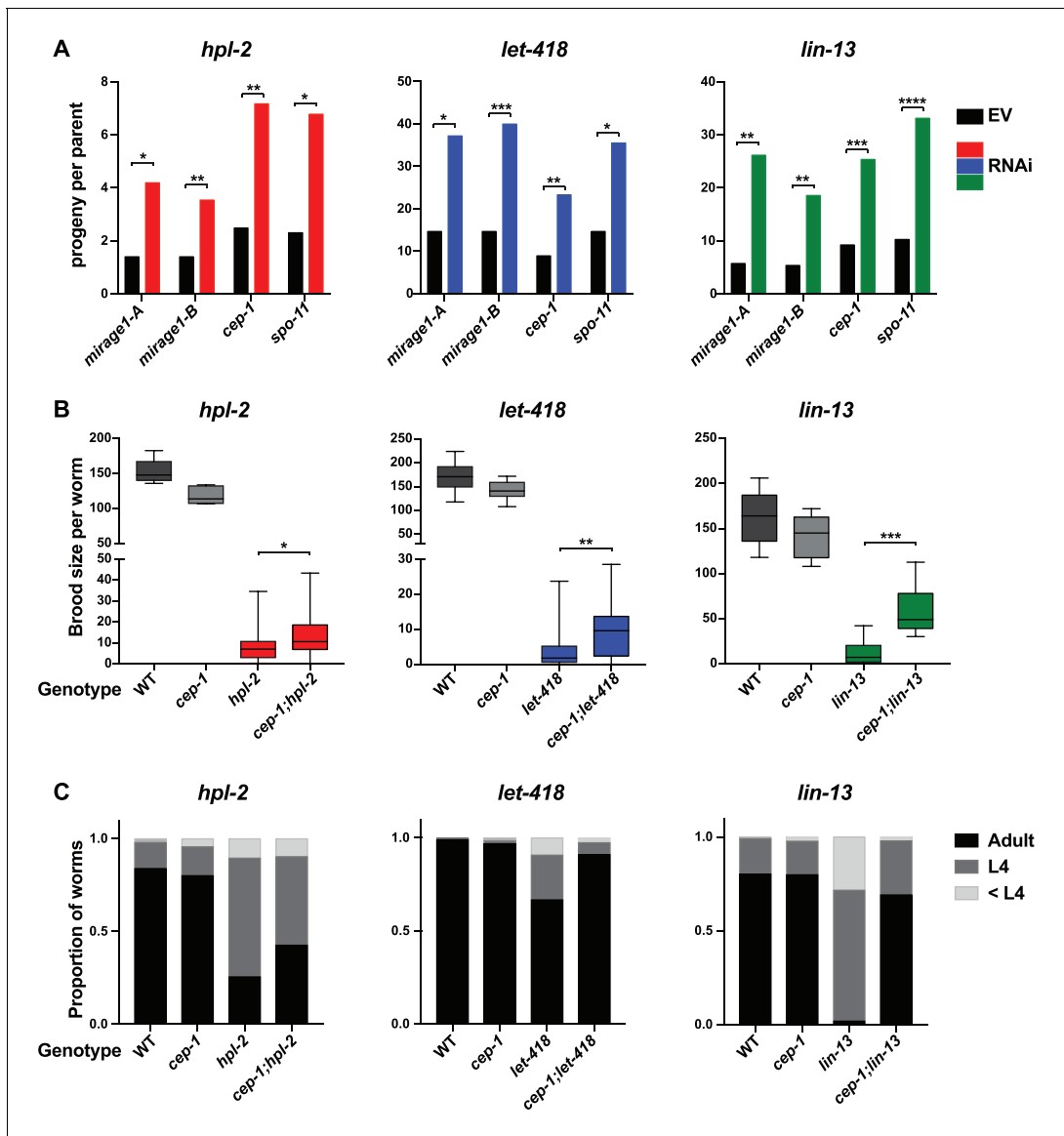

**Figure 4.** Phenotypic suppression of *hpl-2, let-418* and *lin-13* by inhibition of MIRAGE1, *cep-1*/p53, or *spo-11*. (A) RNAi of MIRAGE1, *cep-1*/p53 or *spo-11* partially suppresses sterility of *hpl-2, let-418*, and *lin-13*. Average number of progeny per worm for control empty vector RNAi (EV) or the indicated RNAi treatments in *hpl-2(tm1489)*, *let-418(n3536)*, or *lin-13(n770)* (averages of 5–11 experiments). Experiments were done under conditions where the mutant strain was nearly sterile to detect an increase in fertility (see Materials and methods). Control progeny numbers vary by experiment, but were always paired with experimental RNAi. Stars indicate statistical significance assessed using paired t-tests, comparing experimental to control RNAi ($p < 0.05$, one star; $p < 0.01$, two stars; $p < 0.001$, three stars). Two sets of RNAi clones were used to target MIRAGE1 elements (termed mirage-A and mirage-B). RNAi clones used are given in the methods. (B) Mutation of *cep-1*/p53 partially suppresses sterility of *hpl-2, let-418*, and *lin-13* mutants at 25°C. Statistical significance was assessed using single sided t-tests, asking if *cep-1; hpl-2, cep-1; let-418*, and *cep-1; lin-13* double mutants had larger broods than the corresponding heterochromatin single mutants. See methods for growth conditions. (C) Loss of *cep-1* partially rescues growth delay defect of heterochromatin mutants at 25°C. Developmental stage of worms grown from L1 at 25C for approximately 48 hr was assessed (adult, L4, younger than L4). A representative experiment out of three replicates is shown, assaying between 95 and 213 worms in each. See methods for growth conditions.

which heterochromatin proteins promote normal germ line function is via the repression of repetitive elements, in particular those encoding transposases.

## Heterochromatin mutants display DNA repair defects, increased germ line apoptosis and fertility dependence on CEP-1/p53

Desilencing of repetitive elements has been reported to cause DNA damage in other organisms (*Wallace et al., 2013*); therefore, repeat expression in heterochromatin mutants might lead to genotoxic stress and genome instability in the germ line. Consistent with this, loss of *lin-61* leads to replication stress and genome instability, with increased germ line and somatic mutation frequency (*Johnson et al., 2013*; *Pothof et al., 2003*). *lin-61* mutants also have defects in DNA repair (*Johnson et al., 2013*; *Pothof et al., 2003*). A previous study reported that an *hpl-2(tm1489)* null mutant strain was hypersensitive to ionizing radiation (IR) (*Luijsterburg et al., 2009*); however, we found that this strain also harboured a deletion in the *polq-1* gene, which encodes DNA polymerase theta. Because *polq-1* mutants are reported to have increased sensitivity to DNA damaging agents and display genome instability (*Muzzini et al., 2008*), it was unclear if the defects observed were due to *hpl-2* (see Materials and methods).

To determine whether *hpl-2* has a role in DNA repair, we tested the response of the isolated *hpl-2(tm1489)* mutant to IR induced DNA damage. Following IR, we observed that *hpl-2* mutants show higher levels of oocyte fragmentation compared to wild type, suggesting that they are defective in DNA repair (*Figure 5—figure supplement 1*). Additionally, *hpl-2* mutant germ lines are hypersensitive to induction of phosphorylation of the DNA damage checkpoint kinase CHK-1 (*Figure 5—figure supplement 1*). The hypersensitivity of *hpl-2* and *lin-61* mutants to exogenous DNA damage are consistent with increased genotoxic stress in the germ line.

We considered that the repeat desilencing and DNA repair defects of heterochromatin mutants might lead to increased germ line apoptosis, and thereby contribute to germ line and fertility defects. In wild-type animals, physiological apoptosis occurs in the pachytene region of the gonad, with around half of the initially produced germ cells eliminated by apoptosis as a quality control mechanism (*Gartner et al., 2008*). DNA damage causes increased apoptosis over physiological levels, and this increase is dependent on *cep-1*/p53 (*Schumacher et al., 2001*; *Derry et al., 2001*). To assess germ line cell death in the heterochromatin mutants we used a CED-1::GFP reporter, which allows visualization of apoptotic germ cells in adult animals (*Zhou et al., 2001*). We observed that *hpl-2*, *lin-13*, *lin-61*, and *set-25 met-2* mutants all displayed increased germ line apoptosis (*Figure 5*; *let-418* was not assayed because the apoptosis reporter used is genetically linked). Thus, heterochromatin factor mutants have increased germ cell death.

p53 is important for transduction of the DNA damage response and other stresses, and *C. elegans cep-1*/p53 is required for damage induced cell death (*Schumacher et al., 2001*; *Derry et al., 2001*). We used RNAi to test whether activation of p53 dependent pathways played a role in heterochromatin mutant sterility. Following RNAi of *cep-1*/p53, we found that the brood sizes of *hpl-2*, *lin-13*, and *let-418* mutants grown at 25°C were modestly increased (*Figure 4A*). We further tested the effect of loss of *cep-1* by making double mutants with *cep-1(lg12501)*. Similar to the RNAi results, we observed that mutation of *cep-1* increased the fertility of *lin-13*, *let-418* and *hpl-2* mutants (*Figure 4B*). This increase is not due to a general effect of *cep-1* on fertility, as *cep-1* mutants have a slightly reduced brood size compared to wild-type animals (*Figure 4B*). We also observed that *cep-1* loss partially rescued the slow growth phenotype of the mutants (*Figure 4C*). These results suggest that genotoxic stress and DNA damage signalling in heterochromatin mutants activates p53, which contributes to sterility and slow growth. The increase in fertility upon *cep-1*/p53 inhibition may be a direct consequence of reduced germ line apoptosis, or alternatively the effect may be indirect, by preventing DNA damage signalling or improving growth rate. We also note that although fertility of heterochromatin mutants is increased when *cep-1*/p53 is inhibited, it is not restored to wild-type levels indicating that other mechanisms contribute to sterility.

## SPO-11 induced endogenous DNA damage contributes to heterochromatin mutant sterility

We next investigated whether endogenous physiological DNA damage may also contribute to heterochromatin mutant sterility. During meiosis, double strand breaks are induced by the

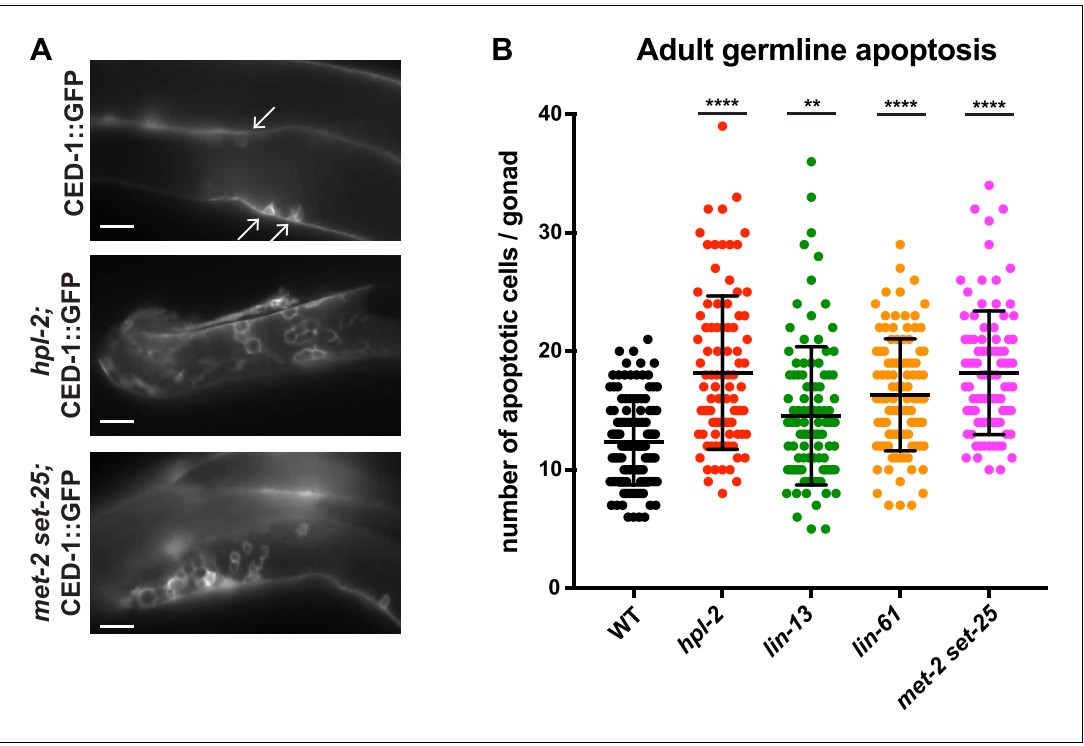

**Figure 5.** Heterochromatin mutants have increased germline apoptosis. CED-1::GFP (*bcIs39* [P*lim-7::ced-1::gfp*]), expressed in gonad sheath cells, marks engulfed apoptotic cells in the pachytene region of the adult gonad. (**A**) CED-1::GFP images for wild type, *hpl-2*, and *met-2 set-25* mutant gonads. Arrows point to engulfed apoptotic cells; scale bar = 16 um. (**B**) Number of apoptotic cells per gonad arm for wild-type (*bcIs39*), *hpl-2(tm1489); bcIs39*, *lin-13(n770); bcIs39*, and *met-2(n4256) set-25(n5021); bcIs39*. Shown are the combined data points of at least three independent replicates; each dot represents an individual gonad arm count. Bars denote mean and SD. Mann-whitney non-parametric tests were performed on mutant versus control. (p-values for *hpl-2, lin-61, met-2 set-25* are <0.0001; p-value for *lin-13* is 0.009). Strains were cultured at 20°C and scored 48 hr after the L4 stage.
*Figure 5—figure supplement 1* shows increased sensitivity of *hpl-2* to IR-induced DNA damage.
The following figure supplement is available for figure 5:

**Figure supplement 1.** *hpl-2* mutants are hypersensitive to ionizing radiation-induced DNA damage.

topoisomerase-like protein SPO-11 to facilitate crossover formation and meiotic recombination (*Dernburg et al., 1998*). Similar to inhibition of damage induced cell death or MIRAGE1 expression, we found that inhibiting meiotic double strand breaks by RNAi of *spo-11* increased the brood size of *hpl-2*, *lin-13*, and *let-418* mutants (*Figure 4A*), suggesting that defects in repair of meiotic double strand breaks contributes to sterility.

## The piRNA pathway shows similarity in repeat regulation and functional connections to heterochromatin factors

The piRNA pathway has a well-known role in preventing the activity of transposons in the germ line (*Weick and Miska, 2014*). In *C. elegans*, the piRNA pathway operates through the Piwi Argonaut protein PRG-1. Silencing occurs both transcriptionally, through engagement of the nuclear RNAi pathway, and post-transcriptionally, through a poorly understood mechanism. Interestingly, *prg-1* mutants have fertility defects, displaying a low brood size and a mortal germline phenotype that is more pronounced at elevated temperatures (*Das et al., 2008*; *Batista et al., 2008*). The observation that heterochromatin mutants desilence repetitive elements together with the finding that HPL-2 and H3K9 methyltransferse SET-25 are needed for piRNA pathway function in conjunction with the

nuclear RNAi pathway (*Ashe et al., 2012*) prompted us to further investigate connections between these factors.

We first assayed the expression of repetitive DNA in *prg-1* mutant adults because genome-wide profiling had not previously been done. We detected upregulation of 18 repetitive elements in *prg-1* mutants, 14 of which are also upregulated in at least one of the heterochromatin mutants, including MIRAGE1 elements (*Figure 6A*, *Figure 3—source data 1* and *2*). RNA FISH experiments showed that MIRAGE1 RNA is increased in *prg-1* mutant germ lines, similar to observations in heterochromatin mutants described above (*Figure 3E* and *Figure 3—figure supplement 3*). Given this overlap in targets and the fertility defects of *prg-1* mutants, we assessed whether they also showed increased germ line apoptosis as seen in heterochromatin factor mutants. Indeed, we observed significantly increased germ cell death in *prg-1* mutant adults (*Figure 6B*). Therefore, the piRNA pathway and heterochromatin factors have shared targets and phenotypes, and likely collaborate in maintaining genomic integrity of the developing germline.

We next used a piRNA activity sensor to test whether heterochromatin factors other than *hpl-2* are needed for piRNA pathway function. Similar to *hpl-2* and *set-25*, we found that *lin-61* and *let-418* mutants derepress the piRNA sensor reporter (*Figure 6C,D*). We also observed weak desilencing in a fraction of *met-2* mutants (*Figure 6D*), which was not observed in a previous assay (*Ashe et al., 2012*). However, the piRNA sensor was not desilenced in *lin-13* mutants. It is possible that the lack of desilencing is due to the *lin-13(n770)* allele being non-null, however this mutant does show defects such as upregulation of repetitive elements and increased apoptosis.

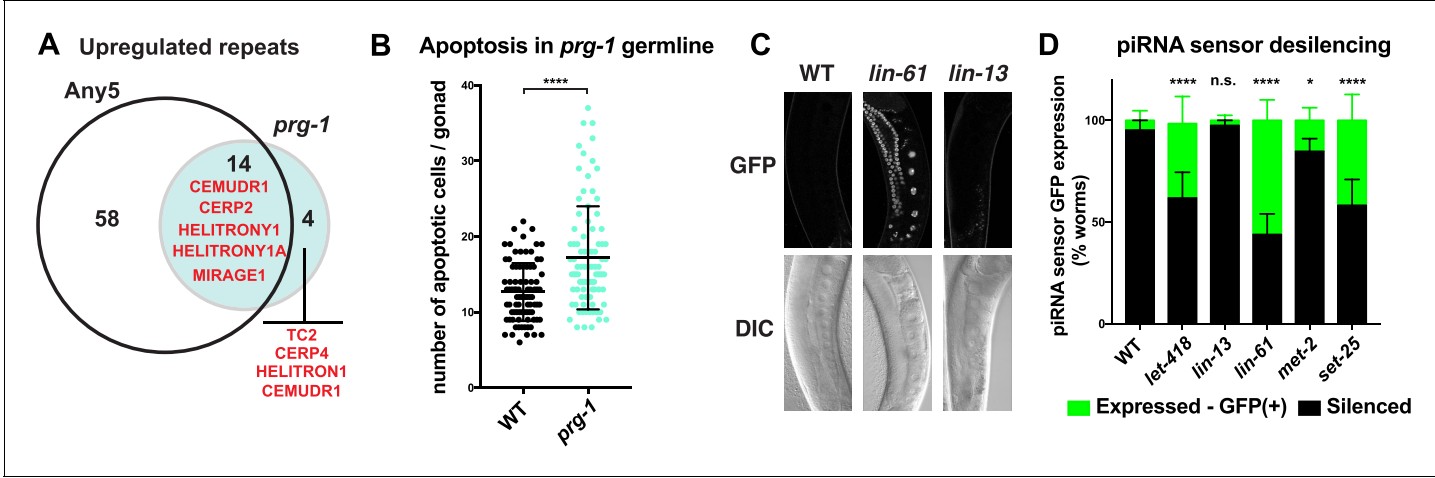

**Figure 6.** Heterochromatin factors interact with the piRNA pathway. (**A**) Venn diagram showing extent of overlap between repeats upregulated in *prg-1* mutants and repeats upregulated in any of the five heterochromatin factor mutant strains (*hpl-2, let-418, lin-13, lin-61,* or *met-2 set-25*). Listed in the Venn are the numbers of repeats and repeat families common or unique to *prg-1*. (**B**) *prg-1* mutant germ lines show increased germ cell death. Shown are the number of apoptotic cells per gonad arm for *bcIs39* (CED-1::GFP) and *prg-1; bcIs39* (CED-1::GFP). Each dot represents an individual gonad arm count. Bars denote mean and SD. A minimum of 25 gonads were scored per experiment and shown are the combined datapoints of at least three independent replicates. Mann-whitney non-parametric tests were performed on mutant versus control (p-value<0.0001). Strains were cultured at 20°C until L4 stage, then shifted to 25°C for 48 hr before scoring. (**C, D**) Heterochromatin mutants desilence a piRNA sensor. piRNA sensor expression (*mjIs144* [mex-5p::HIS-58::GFP::piRNA(21UR-1)::tbb-2–3'UTR]) was quantified in one day old wild-type and heterochromatin mutants cultured at 20°C. (**C**) Representative GFP and DIC microscope images of adult germlines in which the reporter is silent (WT, *lin-13*) or expressed (*lin-61*). (**D**) Quantification of piRNA sensor expression in wild type and heterochromatin mutants. Shown are the means and standard error of the percentage of worms which at least weakly desilenced the GFP reporter in oocytes and pachytene regions. A minimum of 100 worms for each strain was assessed over four independent experiments. Fishers exact tests were performed on the combined datapoints to address significance, with *let-418* (p-value<0.0001), *lin-61* (p-value<0.0001) and *set-25* (p-value<0.0001) all displaying increased frequency of expression of the piRNA sensor reporter, while sensor expression in *lin-13* is not significantly different from wild type (p-value 0.4419). *met-2* mutants weakly desilence the sensor in a small subset of adults scored (p-value 0.0215).

The following figure supplement is available for figure 6:

**Figure supplement 1.** Quantification of piRNAs and dependent 22G RNAs in *prg-1* and *hpl-2* mutants.

A previous study profiling small RNAs in *hpl-2* mutants in a piRNA sensor background showed that piRNAs targeting the sensor or a few endogenous targets were not altered in abundance, suggesting that *hpl-2* acts downstream of piRNA production (*Ashe et al., 2012*). To investigate this further, we compared the global abundance of piRNAs in *hpl-2* and wild-type adults. Similar to the above results, we found that *hpl-2* mutants make normal levels of piRNAs (*Figure 6—figure supplement 1*). We also investigated the production of secondary 22G siRNAs in *hpl-2* mutants. We detected a decrease in 22G RNAs mapping near predicted piRNA target sites in *prg-1* mutants as previously observed (*Lee et al., 2012*), but levels were normal in *hpl-2* mutants (*Figure 6—figure supplement 1*). *hpl-2* mutants also showed normal levels of 22Gs at repeat elements. (*Figure 6—figure supplement 1*). Therefore, at least for *hpl-2*, the role in the piRNA pathway appears to be downstream of piRNA and subsequent 22G RNA synthesis.

To summarize, *hpl-2*, *lin-61*, *let-418*, *set-25*, and *met-2* are important for piRNA pathway function. Nevertheless, the widespread binding sites and desilencing of additional targets relative to *prg-1* indicate that heterochromatin proteins also mediate repression that is not piRNA-induced.

## Partial redundancy between *let-418*/Mi-2 and the nuclear RNAi pathway

The *C. elegans* nuclear RNAi pathway (called the nrde pathway) mediates transcriptional repression and directs H3K9me3 methylation to its targets (*Guang et al., 2010*; *Gu et al., 2012*; *Burton et al., 2011*; *Burkhart et al., 2011*; *Buckley et al., 2012*; *Guang et al., 2008*). The nrde pathway also functions in repression of piRNA targets (*Ashe et al., 2012*). To investigate the relationship between the nrde pathway, the piRNA pathway, and heterochromatin factors in repetitive element regulation, we carried out RNA-seq on *nrde-2(gg91)*, a putative null mutant, and compared results to those of heterochromatin and *prg-1* mutants. We observed that *nrde-2* mutants showed a larger and different spectrum of repetitive element desilencing compared to *prg-1* or any of the heterochromatin mutants (*Figure 7—figure supplement 1*, *Figure 3—source data 1* and *2*). Of 71 elements desilenced in *nrde-2* mutants, only seven overlap a repeat desilenced in one of the heterochromatin mutant strains (*Figure 7—figure supplement 1*). Notably MIRAGE1 elements, prominently upregulated in heterochromatin and *prg-1* mutants, are not desilenced in *nrde-2* mutants (*Figure 3—source data 1* and *2*; *Figure 7—figure supplement 1*). Retrotransposons are highly enriched among *nrde-2* targets (45/71) whereas heterochromatin factors and *prg-1* are more associated with DNA transposon misregulation (*Figure 3B*). Therefore, although the nrde pathway is required for aspects of piRNA function, repetitive element targets largely differ. Notably, the finding that elements derepressed in *nrde-2* mutants differ from those in *met-2 set-25* mutants, which lack detectable H3K9 methylation, suggests that H3K9 methylation may not be required for nrde dependent repression.

Like the heterochromatin factor and *prg-1* mutants, *nrde-2* mutants also show a temperature sensitive decrease in fertility (*Guang et al., 2010*). To test whether the *nrde-2* fertility function had functional overlap with heterochromatin factors, we constructed double mutants between *nrde-2* and three mutants (*hpl-2*, *lin-61*, and *let-418*). We observed no reduction in fertility for *nrde-2; hpl-2*, and a weak but non-significant reduction for *nrde-2; lin-61* double mutants (*Figure 7A*). However, *nrde-2; let-418*, double mutants showed a significantly smaller brood size than expected compared to the single mutants, indicating partial functional redundancy between *let-418* and the nrde pathway (*Figure 7A*). We also observed that all three double mutants showed significantly increased embryo lethality compared to the single mutants (*Figure 7B*).

To further investigate the genetic interaction between *nrde-2* and *let-418*, we carried out RNA-seq of *nrde-2; let-418* adults to test for redundancy in repeat element repression. We found that more repeats are upregulated in *nrde-2; let-418* double mutants compared to *nrde-2* or *let-418* single mutants (*Figure 7B*). Most of the repeat elements upregulated only in the *nrde-2; let-418* double mutant are retrotransposons (27/46; *Figure 7B* and *Figure 3—source data 2*). We conclude that NRDE-2 and LET-418 have unique and redundant roles in the repression of repetitive DNA elements.

Because a key output of the nrde pathway is the deposition of H3K9me3, the observed redundancy between NRDE-2 and LET-418 prompted us to investigate whether the nrde pathway might control H3K9me3 levels at heterochromatin regulated loci. To this end, we used published H3K9me3 ChIP-seq datasets in four different nrde mutants (*hrde-1, nrde-2, nrde-3,* and *nrde-4*; (*Gu et al., 2012*; *Buckley et al., 2012*; *Ni et al., 2014*). and analysed levels at genes and repeats

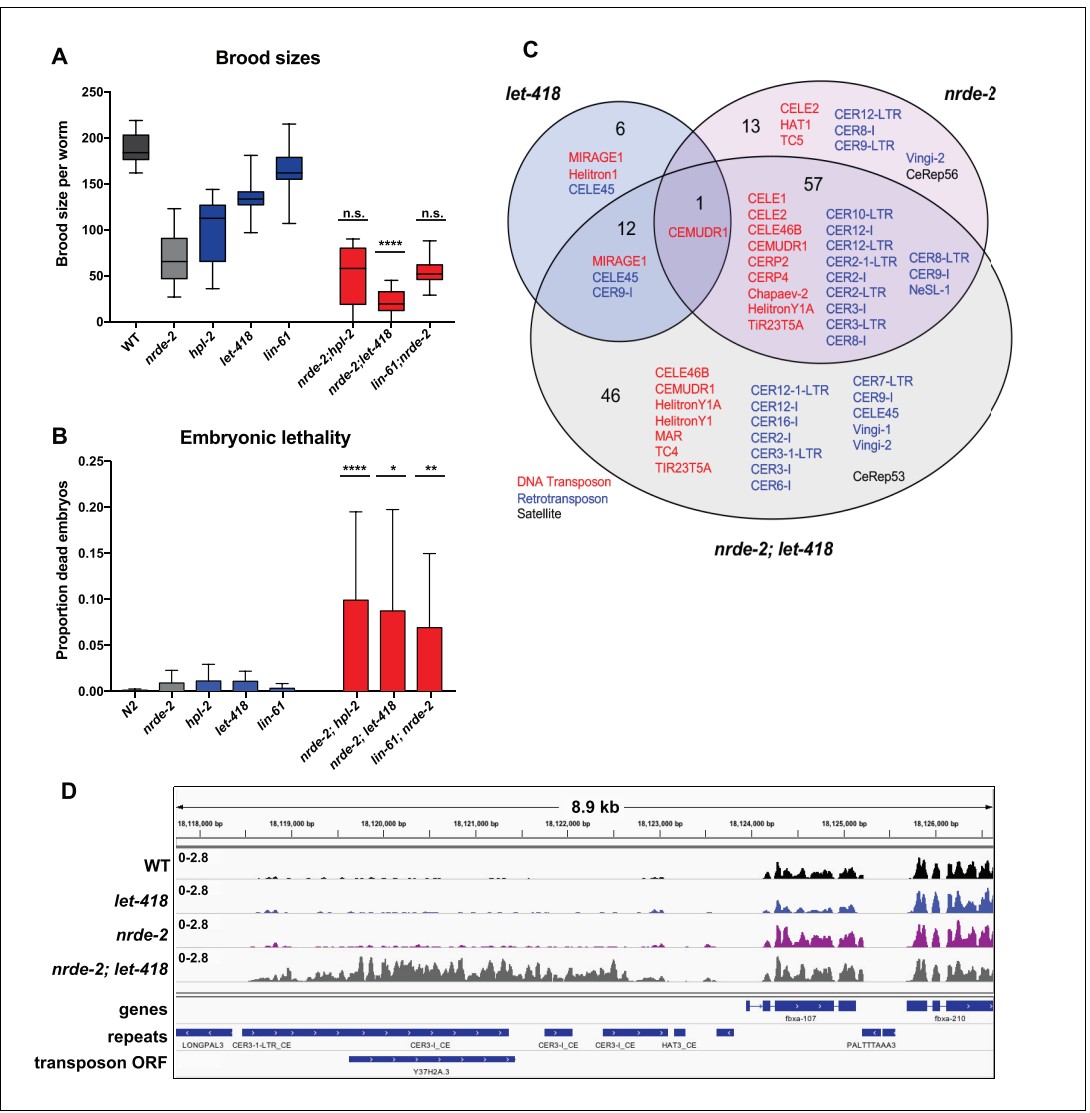

**Figure 7.** *nrde-2* and *let-418* show functional redundancy. (**A**) *nrde-2* and *let-418* mutants show genetic interaction in fertility. Brood sizes of *nrde-2; hpl-2, nrde-2; let-418*, and *nrde-2; lin-61* double mutants were compared to those of single mutants. Synchronized single or double mutant strains of the indicated genotype were grown at 15°C until the L3 stage and then transferred to 25°C, and total progeny including dead embryos determined for 12–24 mothers across two independent experiments. A single-sided Mann-Whitney U test was used to determine whether the double mutant had a lower brood size than expected under a multiplicative model of interaction when compared to the individual single mutants. Brood size is significantly lower than expected for *nrde-2; let-418* (p=9.21E-11) but not *lin-61; nrde-2* (p=0.20) or *nrde-2; hpl-2* (p=98). (**B**) *nrde-2; hpl-2, nrde-2;let-418* and *lin-61; nrde-2* double mutants show increased proportion of dead embryos within their broods compared to single mutants. Total number of dead embryos was determined as a proportion relative to their total brood size for the experiment in (**A**). Mann-Whitney U tests were performed to compare the proportion of unhatched eggs in double mutants relative to *nrde-2* single mutants, and were all found to be significant at p<0.05. (**C**) Repeat families with members upregulated in *let-418*, *nrde-2*, and *nrde-2; let-418* young adult worms. *Figure 7—figure supplement 1A* compares repeat families upregulated in *nrde-2, prg-1,* or any of the five heterochromatin mutants (**D**) Example of repeats upregulated in *nrde-2; let-418*, but not the single mutants. Tracks are RNA-seq reads per million of two combined replicates. *Figure 7—figure supplement 1B* shows lack of MIRAGE1 expression in the *nrde-2* mutant background.

The following figure supplements are available for figure 7:

**Figure supplement 1.** Overlap of *nrde-2, prg-1*, and heterochromatin targets.

*Figure 7 continued on next page*

*Figure 7 continued*

**Figure supplement 2.** H3K9me3 levels on repeats in nrde mutants.
**Figure supplement 3.** H3K9me3 levels on genes in nrde mutants.

upregulated in *nrde-2* mutants or only in heterochromatin mutants. The nrde pathway acts in the germ line and soma: HRDE-1 and NRDE-3 are argonautes specific for germ line or soma, respectively, whereas NRDE-2 and NRDE-4 act in all tissues (*Guang et al., 2010*; *Burkhart et al., 2011*; *Buckley et al., 2012*; *Guang et al., 2008*). We observed that repeat elements and genes upregulated in *nrde-2* mutants also have reduced H3K9me3 levels, supporting the link between H3K9me3 methylation and repression of endogenous targets (*Figure 7—figure supplements 2* and *3*). H3K9me3 was also reduced on *nrde-2* upregulated elements in mutants of other nrde genes that act in the germ line (*hrde-1* and *nrde-4*), but not in the soma specific argonaute mutant *nrde-3* (*Figure 7—figure supplements 2* and *3*). This suggests that the transcriptional upregulation observed in *nrde-2* mutants occurs largely in the germ line.

We next analysed H3K9me3 levels in sets of genes and repeats upregulated in heterochromatin mutant strains but not in *nrde-2* mutants to ask if these elements were also under nrde control. Indeed, genes upregulated in any of the five heterochromatin factor mutant strains (*hpl-2, lin-13, let-418, lin-61*, *met-2 set-25*) but not in *nrde-2* mutants also showed reduced H3K9me3 in germ line nrde mutants, though the reduction was weaker than for *nrde-2* regulated genes (*Figure 7—figure supplement 3*). Repeats upregulated only in heterochromatin factor mutants showed a trend of reduced H3K9me3 (*Figure 7—figure supplement 2*). Therefore, the germ line nuclear RNAi pathway partially controls H3K9me3 levels at loci regulated by heterochromatin factors. Because these elements are not upregulated in *nrde-2* mutants, this indicates that the observed reduction of H3K9me3 is not sufficient for derepression and supports partial redundancy between the nrde pathway and heterochromatin factors in repeat silencing.

## Discussion

All animal genomes contain abundant repetitive elements, which are subject to silencing control. This study expands our knowledge of repetitive element silencing by showing that a diverse set of heterochromatin factors (HPL-2/HP1, LIN-61, LET-418/Mi-2, LIN-13, and MET-2) work together with the piRNA and nuclear RNAi pathways to silence repetitive elements such as DNA transposons and retrotransposons. The systematic analyses of multiple factors, most of which are conserved, uncovered a network of functional interactions between them. We suggest that the interactions we identify here are likely to be relevant to the control of repetitive elements in other animals.

All factors and pathways studied are individually important for germ line function, as evidenced by reduced fertility or sterility of single mutants, and all are individually necessary for repetitive element silencing. Importantly, functional redundancy among the factors and pathways demonstrates widespread safeguards for ensuring germ line health and fertility. Our results show that there are interacting and overlapping mechanisms of repeat element silencing (*Figure 8A*).

### Heterochromatin factors and small RNA pathways

Connections between heterochromatin formation and transcriptional silencing via RNAi mechanisms involving small RNAs have been observed in a variety of organisms (*Castel and Martienssen, 2013*; *Holoch and Moazed, 2015*; *Martienssen and Moazed, 2015*). For instance, RNAi machinery directs silencing at repetitive centromeric regions in *S. pombe* in a process that involves H3K9 methylation and the chromodomain protein Swi6, which is similar to HP1 (*Holoch and Moazed, 2015*; *Hayashi et al., 2012*; *Motamedi et al., 2008*; *Rougemaille et al., 2012*). A second *S. pombe* HP1 homolog, Chp2, functions in transcriptional repression in heterochromatin downstream of the RNAi factors via a complex called SHREC2, which contains Mit2, an Mi-2 related protein (*Holoch and Moazed, 2015*; *Motamedi et al., 2008*). These two HP1 homologs each make partial contributions to silencing, since swi6 chp2 double mutant cells have a stronger silencing defect than either single mutant, suggesting partial redundancy in the two processes (*Motamedi et al., 2008*). Nuclear

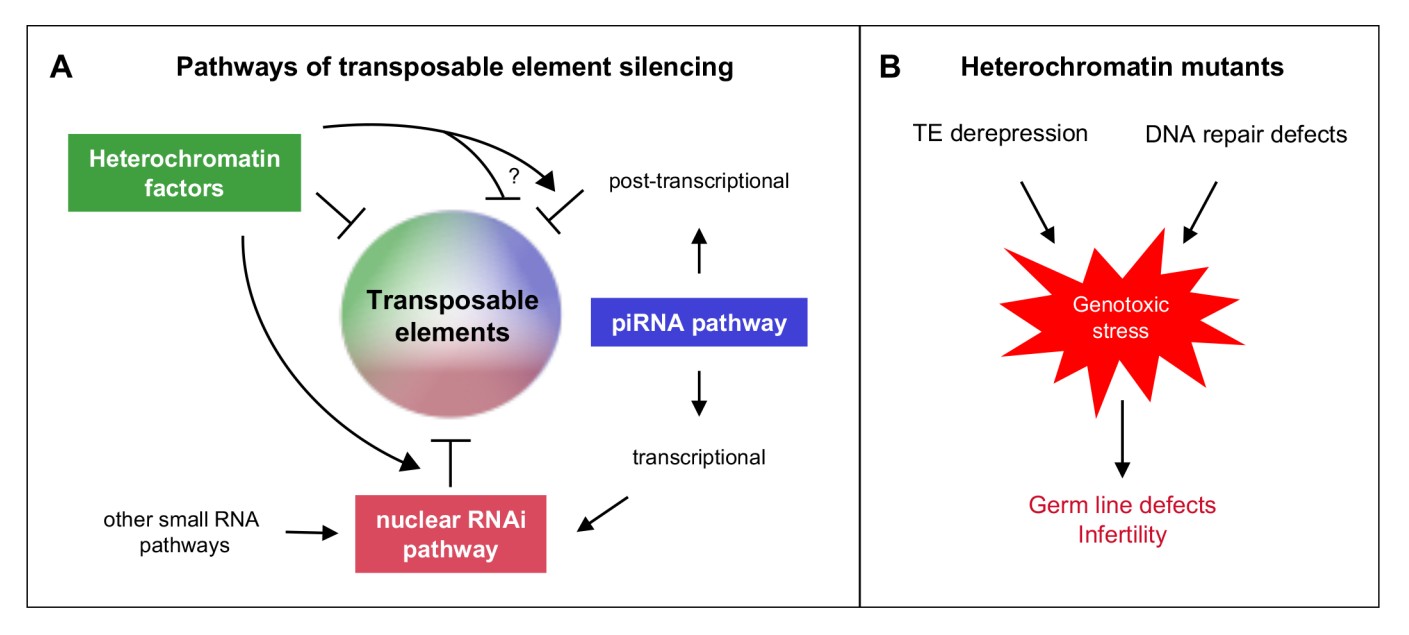

**Figure 8.** Heterochromatin proteins collaborate with small RNAi pathways to maintain fertility. (**A**) Pathways of transposable element silencing in *C. elegans*. Heterochromatin factors participate in repetitive element silencing together with the piRNA and nuclear RNAi pathways, as well as targeting elements independently of these pathways. (**B**) Derepression of transposable elements and defects in DNA repair likely generate genotoxic stress that leads to germ line defects and infertility in heterochromatin factor mutants.

pathways involving small RNAs, H3K9 methylation, HP1 homologs and/or Mi-2 proteins also repress repetitive elements and genes in different eukaryotes including Arabidopsis, Drosophila, mouse, and humans (*Castel and Martienssen, 2013*; *Holoch and Moazed, 2015*; *Martienssen and Moazed, 2015*; *Friedli and Trono, 2015*; *Iwasaki et al., 2015*). However, the mechanisms linking silencing pathways are not fully understood and factors involved in heterochromatin formation and function remain to be identified. For example, a recent study of HP1a interactors in Drosophila identified many new proteins needed for gene silencing and/or heterochromatin organization (*Swenson et al., 2016*).

Previous studies in *C. elegans* have also uncovered connections between heterochromatin factors and small RNA pathways. The piRNA pathway is a small RNA pathway active in the germ line that silences transposons and genes through cytoplasmic and nuclear mechanisms (*Weick and Miska, 2014*). In the nucleus, the piRNA pathway engages the nrde pathway and heterochromatin factors for transcriptional repression: HPL-2, the H3K9me3 histone methyltransferase SET-25, and nuclear RNAi factors are necessary for silencing a piRNA pathway sensor (*Ashe et al., 2012*). The nrde pathway directs H3K9me3 both endogenously and in response to exogenous dsRNA and effects transcriptional repression in both the germ line and soma (*Guang et al., 2010*; *Gu et al., 2012*; *Burkhart et al., 2011*; *Guang et al., 2008*). Endogenous germ line nrde targets have been suggested to include retrotransposons based on their enrichment within genomic intervals that display increased expression and decreased H3K9me3 levels in germ line nuclear RNAi mutants (*Ni et al., 2014*). Here, through genetic and profiling analyses, we have established additional connections between heterochromatin factors and small RNA pathways, linking them to repetitive element repression, uncovering functional redundancy, and expanding understanding of their relationships.

## Heterochromatin factors associate with and repress repetitive elements

We found that the genome-wide distributions of each of the five heterochromatin factors studied here (HPL-2/HP1, LIN-13, LIN-61, LET-418/Mi-2, and MET-2/SETDB1) are highly correlated with each other and strongly associated with repetitive elements. These patterns are correlated with H3K9me2, but not with H3K9me3, as previously seen for HPL-2 in embryos (*Garrigues et al., 2015*).

H3K9me2 and H3K9me3 modifications largely do not overlap (this study and [*Liu et al., 2011*]), but both are associated with repetitive elements. We further found that H3K9me2, but not H3K9me3 is enriched at telomeres. These patterns support functional differences between H3K9me2 and H3K9me3.

In addition to the similarity in binding profiles, we found that *hpl-2, lin-13, lin-61, let-418*, and *met-2 set-25* mutant strains all showed derepression of repetitive elements and genes. While this paper was under review, Zeller et al reported that *met-2 set-25* mutants derepressed transposable element expression in embryos and gonads, similar to the results presented here for *met-2 set-25* young adults (*Zeller et al., 2016*). The global loss of H3K9 methylation also leads to transposon derepression and mobilization in Drosophila (*Penke et al., 2016*). The patterns of upregulated repetitive elements and genes among the five heterochromatin mutant strains are strikingly similar, indicating shared targets. However, although binding profiles and consequences of loss are similar, the factors also have unique roles. Furthermore, genetic interactions between all pairs show that they have partially redundant functions. To understand these relationships, it will be important to investigate their interdependencies and the consequences of inactivating multiple factors together.

## Relationships between heterochromatin factors and small RNA pathways

Comparing expression profiles of heterochromatin mutants to those of *prg-1* (piRNA pathway) and *nrde-2* (nuclear RNAi pathway) mutants, we observed that repeat elements and genes have similar patterns of derepression in heterochromatin factor and *prg-1* mutants, but these largely differ from those in *nrde-2* mutants. Consistent with a functional link between the heterochromatin factors and the piRNA pathway, a previous study showed that HPL-2 and SET-25 are needed for piRNA pathway function (*Ashe et al., 2012*). Here we further found that LIN-61, LET-418, and MET-2 (weakly) are also important. It appears that heterochromatin proteins act as downstream effectors of the piRNA pathway rather than having a role in small RNA biogenesis or stability since levels of piRNAs and their secondary 22G RNAs are normal in *hpl-2* mutants (*Ashe et al., 2012*) (and this study). Investigating whether small RNA populations are altered in other heterochromatin mutants will be needed to confirm this hypothesis. This mechanism appears to differ from the situation in *S. pombe*, where H3K9 methylation and the chromodomain protein Swi6 are required for the association of silencing complexes that generate siRNAs (*Holoch and Moazed, 2015*; *Martienssen and Moazed, 2015*; *Motamedi et al., 2008*, *2004*; *Verdel et al., 2004*). However, it is possible that redundancy between heterochromatin factors in *C. elegans* may have masked involvement in small RNA production.

Many more elements are desilenced in *nrde-2* mutants than in *prg-1* mutants, with little overlap between the two. The elements derepressed in *nrde-2* mutants are mostly LTR retrotransposons, in line with a study finding transcriptionally upregulated genomic intervals in *hrde*-1 mutants to be enriched for LTRs and unaffected in *prg-1* mutants (*Ni et al., 2014*). The apparent difference in targets between NRDE-2 and PRG-1 could be due to the requirement for PRG-1 in initiation but not maintenance of silencing. Once silencing is established by PRG-1, nuclear RNAi maintains silencing in a process termed RNAe that depends on continued generation of secondary siRNAs by mutator proteins, the presence of secondary siRNA-associated argonautes (including *nrde-2*), and maintenance of the established chromatin state by heterochromatin factors (*Ashe et al., 2012*; *Shirayama et al., 2012*; *Luteijn et al., 2012*; *Lee et al., 2012*; *de Albuquerque et al., 2015*).

Our study also uncovered interactions between heterochromatin factors and the nuclear RNAi pathway. We found that *let-418; nrde-2* double mutants show a strongly enhanced fertility defect compared to the single mutants, and that they desilence a larger set and a wider spectrum of repetitive elements. Therefore, for some elements, either LET-418 or NRDE-2 is sufficient for silencing, demonstrating redundancy in repetitive element silencing. These interactions further emphasize the overlapping safeguards that function to effectively repress repetitive elements. The increased embryo lethality seen in double mutants between *nrde-2* and three tested heterochromatin factors (*hpl-2, lin-61*, or *let-418*) suggest additional as yet unexplored redundancy between the nuclear RNAi pathway and heterochromatin factors. That a substantial number of repetitive elements are desilenced in heterochromatin factor mutants but not in *prg-1* or *nrde-2* mutants suggests that heterochromatin factors can also act as independent agents, silencing repetitive elements independently of small RNA pathways.

Interestingly, genes and repeats upregulated in *nrde-2* mutants more often have high levels of H3K9me3 than H3K9me2 whereas those upregulated in *met-2 set-25* mutants (which lack all H3K9 methylation) show the opposite pattern and more often have high H3K9me2 marking (*Figure 3—figure supplement 2*). The association between H3K9me3 and *nrde-2* upregulated genes suggests that the nuclear RNAi pathway may specifically engage this modification. Consistent with this, repeats and genes upregulated in *nrde-2* mutants have strongly reduced H3K9me3 levels in germ line nuclear RNAi pathway mutants. However, the difference in elements desilenced in *nrde-2* and *met-2 set-25* mutants (which lack H3K9 methylation) argues against an essential requirement for H3K9 methylation in nrde mediated repression. We observed that H3K9me3 levels in *nrde-2* and other germline nuclear RNAi mutants are also weakly reduced at genes and repeats repressed by heterochromatin pathway factors, even though these elements are not upregulated in *nrde-2* mutants. This suggests that the heterochromatin factors and the nuclear RNAi pathway may regulate many common elements, but that heterochromatin factors can still effectively silence them in the absence of the nuclear RNAi pathway. Future analyses in mutants compromised for both nuclear RNAi and heterochromatin factors will be needed to address the mechanisms of this redundancy.

## Heterochromatin factors may act locally within euchromatic domains

We speculate that in many cases, repetitive element regulation involves a local mechanism rather than the spreading of large heterochromatin domains. First, we observe that heterochromatin factor binding is often closely associated with repetitive elements and does not extend to adjacent genes. Second, genes containing repetitive elements bound by heterochromatin factors (usually within introns) are often expressed. Indeed, one fourth of genes contain a repetitive element bound by a heterochromatin factor, and of those, 59% (3287/5568) are expressed in the germ line. Formation of a large inactive heterochromatin domain would clearly be incompatible with such widespread germ line expression.

In *C. elegans*, most H3K9 marking occurs on the distal chromosome arms, where small regions of H3K9 are interspersed with chromatin typical of euchromatin (*Liu et al., 2011*). Many active genes, including those expressed in the germ line, reside in these arm regions (*Liu et al., 2011*). Intriguingly, the repeat- and H3K9-rich chromosome arms are generally associated with the nuclear lamina, a region implicated in transcriptional repression (*Towbin et al., 2012*; *Ikegami et al., 2010*). Active genes at the periphery that contain repetitive elements bound by heterochromatin factors may be subject to special mechanisms for their expression.

## Heterochromatin factors and DNA repair

In addition to functioning in repetitive element repression, some of the heterochromatin factors we studied here are implicated in DNA repair or genome stability. We found that *hpl-2* mutant germ lines show reduced repair and increased activation of DNA damage signaling in response to ionizing radiation. LIN-61 is needed for DNA repair in the germ line and its loss causes an increase in the germ line mutation rate (*Johnson et al., 2013*; *Pothof et al., 2003*). Additionally, both *lin-61* and *set-25* were identified in a genome-wide RNAi screen for genes needed for genome stability in the soma (*Pothof et al., 2003*). Furthermore, *met-2 set-25* mutants, which lack all H3K9methylation, were recently shown to have increased sensitivity to replication stress and increased rates of repeat associated mutations and R loops (*Zeller et al., 2016*). In mammalian cells, orthologs of LET-418 (Mi-2), HPL-2 (HP1), and MET-2 (SETDB1), have documented roles in DNA repair (*Alagoz et al., 2015*; *Polo et al., 2010*; *Urquhart et al., 2011*). Mi-2 is needed for recruitment of DNA repair proteins to sites of DSBs, and loss of Mi-2 from human fibroblast cell lines leads to apoptosis and sensitivity to ionizing radiation (*Larsen et al., 2010*; *Pan et al., 2012*; *Smeenk et al., 2010*). HP1 accumulates at DSBs and its depletion causes abnormal recruitment of repair factors (*Soria and Almouzni, 2013*). Additionally, DNA damage repair defects caused by depletion of SETDB1 are similar to those seen upon loss of HP1 (*Alagoz et al., 2015*). Defects in repair of DNA lesions are likely to cause germ line stress and to contribute to the germ line instability, germ line development defects, and increased germ line apoptosis seen in heterochromatin factor mutants. Such processes might also underlie the reduced fertility and increased germ line apoptosis seen in *prg-1* mutants.

The activation of MIRAGE1 DNA transposases in all heterochromatin factor mutant strains and *prg-1* mutants would be expected to cause double strand breaks and/or replication stress. We show

that this abnormal expression contributes to sterility because fertility was partially restored in *hpl-2*, *let-418*, and *lin-13* mutants when MIRAGE1 transcripts were knocked down via RNAi. Similarly, inhibiting endogenous meiotic double strand breaks by knockdown of *spo-11* also partially restored fertility. These results suggest that heterochromatin factors act to combat different types of genotoxic insults, both through silencing repetitive elements and facilitating repair. If not dealt with, these insults cause sterility (*Figure 8B*).

We also observed that loss of *cep-1*/p53 suppressed heterochromatin factor defects. p53 is important for mediating DNA damage signaling (*Meek, 2009*). In *C. elegans*, p53/*cep-1* is required for damage induced germ cell death (*Schumacher et al., 2001*; *Derry et al., 2001*). In the soma, DNA damage signaling does not lead to p53/CEP-1-mediated apoptosis; however, CEP-1 does play a role in DNA repair in the soma, and it slows larval development in response to loss of CLK-2/TEL2 DNA damage signaling (*Derry et al., 2007*; *Hoffman et al., 2014*). Our findings that loss of *cep-1* partially suppresses the sterility and slow growth phenotypes of *hpl-2, lin-13*, and *let-418* suggests that damage signaling and *cep-1*/p53 underlies these defects.

Interestingly, none of the heterochromatin mutant strains studied here display hallmarks of mutators such as high embryo lethality or the frequent production of progeny with visible mutant phenotypes. We propose that quality control mechanisms in the germ lines of heterochromatin factor mutants largely prevent improperly repaired meiotic germ cells from becoming mature gametes, either through apoptosis or the arrest of gametogenesis, to ensure that mutation rates are low.

## Conclusions

This study indicates a complex orchestration of fertility protection by HPL-2/HP1, LIN-13, LIN-61, MET-2, and LET-418/Mi-2 together with the piRNA and nuclear RNAi pathways. Repression of repetitive elements may prevent replication stress and DNA damage, but when damage does occur, these heterochromatin proteins participate in repair pathways to maintain genome integrity. Further work will be required to untangle the mechanisms and individual roles in DNA repair pathways and repetitive element repression.

## Materials and methods

### Worm culture and strains

Strains were cultured using standard methods (*Brenner, 1974*). Strains used in the paper are given in *Supplementary file 2*. Whole genome sequencing of PFR40 *hpl-2(tm1489)* identified an 882 bp deletion in the *polq-1* gene at chrIII:5792238–5793119. The underlined T residue marks the junction of the deletion and matches the flanking sequence of both sides of the deletion: TAAATCTCTA TCCGATGTGATCCACGTCGATAACATTATTC; we have named this lesion *polq-1(we100)*. The JA1902 strain harboring *hpl-2(tm1489)* but lacking *polq-1(we100)* was derived by outcrossing MT15062 *hpl-2(tm1489);hpl-1(n4317)*, which does not contain *polq-1(we100)*.

### ChIP-seq

Wild-type young adults (YA) were prepared by growing synchronized L1s in liquid culture using standard S-basal medium with HB101 E. coli for 60 hr at 20°C. Adults were sucrose floated, washed in PBS, and flash frozen in liquid nitrogen. Extract preparation and chromatin immunoprecipitation were performed as in *Kolasinska-Zwierz et al. (2009)* with the following modifications: tissue was fixed for 10 min in 1.5 mM EGS (Pierce 21565) then formaldehyde added to 1% for a further 10 min before quenching with 0.125M glycine and washing 2X with PBS plus protease inhibitors. Pellets were washed once in FA buffer, then resuspended in 1 ml FA buffer per 1 mL of ground worm powder and the extract sonicated to an average size of 250 base pairs with a Diagenode Bioruptor or Bioruptor Pico for 28 pulses of 30 s followed by 30 s pause. Antibodies used for ChIP are given in *Supplementary file 3*. Following ChIP and DNA purification, libraries were prepared using the Illumina TruSeq kit. Fragments in the 250–300 base pair range were selected using Agencourt AMPure XP beads. Two biological replicate ChIPs were conducted for each factor.

## RNA-seq

Synchronized, starved L1 stage worms were grown on NGM plates under one of two conditions. Condition 1 (*hpl-2*, *let-418*, *lin-61*, *met-2 set-25*, and N2): growth was at 20°C until the L4 stage and then worms were shifted to 25°C for 15–18 hr until they reached young adult stage. Condition 2 (*lin-13*, *prg-1*, *nrde-2*, *nrde-2; let-418,* and N2): growth was at 15°C until the L4 stage and then worms were shifted to 25°C for 15–18 hr until they reached young adult stage. Worms were then harvested, flash frozen in liquid nitrogen, and stored at −80°C until use. RNA was extracted from frozen worms using TriPure (Roche). RNA was purified with Zymo Research RNA Clean and Concentrator-5 (Cambridge Bioscience) following DNAse I digestion. Ribosomal RNA was depleted using Ribo-Zero rRNA Removal Kit (Human/Mouse/Rat) (Illumina). Libraries were prepared using the NEBNext Ultra Directional RNA Library Prep Kit for Illumina (New England Biolabs). Two biological replicates were prepared for each strain.

## Data processing

ChIP-seq and RNA-seq libraries were sequenced using Illumina HiSeq. Reads were aligned to the WS220/ce10 assembly of the *C. elegans* genome using BWA v. 0.7.7 ( *Li and Durbin, 2010*) with default settings (BWA-backtrack algorithm). The SAMtools v. 0.1.19 'view' utility was used to convert the alignments to BAM format. To be able to investigate binding and expression at repetitive elements, we used all aligned reads (mapq0) to generate pileup and normalised tracks. Normalized ChIP-seq coverage tracks were generated using the BEADS algorithm (*Cheung et al., 2011*) without the mappability correction step. ChIP-seq and RNA-seq library read numbers and alignment statistics are given in *Figure 2—source data 3*.

## Peak calls

Broad and sharp ChIP-seq peaks were generated as follows. Initial ChIP-seq peaks were called using MACS v. 2.1.1 (*Feng et al., 2012*) with a permissive 0.7 q-value cutoff and fragment size of 150 bp against a summed ChIP-seq input. These were used in conjunction with a modified IDR procedure to generate broad peak calls ([*Li et al., 2011*]; https://www.encodeproject.org/software/idr/) with an IDR threshold of 0.05 to combine replicates. These broad peaks are termed 'IDR peaks' (*Figure 2—source data 1*). The pipeline for generating IDR peaks is available here: https://github.com/Przemol/biokludge/blob/master/macs2_idr/macs2_idr.ipy. To generate sharp peak calls, the IDR calls were further refined using an adhoc post-processing step, as visually distinct peaks close to each other were often contained within single broad peaks. We identified concave regions within the IDR peaks using the smoothed second derivative of the mapq0 pileup coverage signal with 250 bp kernel (https://github.com/Przemol/biokludge/blob/master/macs2_idr/concave_regions.py). We empirically found the minimum of the second derivative within a concave region to be a good indicator of a visually compelling peak, and used concave regions (within IDR peaks) with a threshold of lower than −500 curvature index. Next, we discarded peaks with MASC2 score lower than 100 and peak width lower than 100 bp. The resulting peaks were filtered against combined ENCODE (http://www.broad-institute.org/~anshul/projects/worm/blacklist/ce10-blacklist.bed.gz) and in-house blacklist (https://gist.github.com/przemol/8a712a2e840f95237f4a4f322f65bee1) to generate our final sharp peak calls, described as 'concave peaks.' We created a summary peak call super set by creating a union of the five heterochromatin factor concave peak calls. We termed this set 'Any5' (n = 33301; *Figure 2—source data 1*). Each Any5 region was then annotated for overlap with each factor. Venn diagrams were plotted using VennDiagram R package (*Chen and Boutros, 2011*), and UpSet plots were generated as described in *Lex et al. (2014)*. For determination of factors bound to repeats, we used broad IDR peak calls since repeats usually display a pattern of broad factor binding. Broad IDR peaks were used in *Figure 3—figure supplement 2*.

## Differential expression analyses of genes

We built an exon model based on Ensembl Gene 77 (Nov 2014) database gene annotation lifted over to ce10/WS220. Tag counts for each gene were extracted from BAM alignment files using HTSeq method working in union mode and implemented in R. These values were used to build an expression matrix. Differential gene expression between N2 and mutant backgrounds was tested using DESeq2; mutants were compared to their temperature matched control N2 replicates

(*Love et al., 2014*). Reads per kilobase of exon model per million mapped reads (RPKM) normalized expression values were generated using the median ratio method (Equation 5 in *Anders and Huber, 2010*). RPKM values, maximum posterior estimates of log2 FC (LFC) and statistical significance estimates for each gene is in *Figure 3—source data 3*. We used a false discovery rate (FDR) < 0.01 and LFC > 1 to call genes up-regulated, and FDR < 0.01 and LFC < −1 to call genes down-regulated. To avoid small differences in developmental stages from contributing to apparent gene expression differences, we also excluded genes whose wild-type expression oscillates repeatedly during development (Supplemental Table S7 in *[Latorre et al., 2015]*).

## Differential expression analyses of repeats

We built a repeat element model based on Dfam 2.0 ([*Hubley et al., 2016*], downloaded Sept 2015 from http://dfam.org/). The model contained 62331 individual repeats divided into 184 families. Since individual repeats did not had unique identifiers (UID), we named them based on genomic position in 'chromosome:start-end' convention, e.g. 'chrI:10773–11032'. Tag counts for each repeat were extracted from BAM alignment files using HTSeq method working in union mode and implemented in R. These values were used to build expression matrixes. Differential repeat expression between N2 and mutant backgrounds was tested using DESeq2 as described above for genes. A table containing RPKM values, maximum posterior estimates of log2 FC (LFC) and statistical significance estimates for each repeat is available in *Figure 3—source data 1*. Upregulated repeats were defined as those with a false discovery rate (FDR) < 0.01, and LFC > 0. In addition, repeats that overlapped a gene upregulated in the matched mutant background were filtered out. For purpose of filtering, differentially expressed genes were defined with more permissive cutoffs: FDR < 0.05 and LFC > 0.

To assess expression of individual repeats scored as upregulated above, we counted uniquely mapping reads, defined as having a BWA mapping quality over 10. Elements with >10 unique reads and fold-change >1.5 were considered upregulated, which applied to 61 of 71 elements upregulated in any of the heterochromatin factor mutant strains. The remaining 10 elements had insufficient uniquely mapping reads for assessment.

## Telomere enrichment

Telomere enrichment for ChIP-seq factors were determined by counting reads with the telomere sequence 'GCCTAA'. Reads were extracted from BAM files (including non- aligned reads) and trimmed to 36 bp. Then the number of 'GCCTAA' motifs was counted for each read using Biostrings R package. Telomeric reads were defined as those having 5 or 6 'GCCTAA' motifs in 36 bp. To assess the statistical significance of enrichment we used one sided Mann–Whitney U test (two replicates for each factor vs. input background of 129 experiments) and reported the p-values.

## Small RNA analyses

The following small RNA datasets from *Ashe et al. (2012)* were used: *prg-1* (GSM708661), WT matching *prg-1* (GSM708660), *hpl-2* (GSM950181), WT matching *hpl-2* (GSM950180), *nrde-2* (GSM950179), WT matching *nrde-2* (GSM950178). Uniquely matching positions in each dataset were determined and the smallest number (530039, in the *prg-1* dataset) subsampled from each. piRNA number was then determined by calculating the number matching the piRNAs annotated in *Batista et al. (2008)* or *Weick and Miska (2014)* (n = 27884 piRNAs). piRNA targets were determined as in *Lee et al. (2012)*, requiring a perfect match and no more than one G:U pair in the seed region (nt 2–8), and allowing up to two mismatches and an additional G:U pair outside of the seed region, excluding self hits (n = 391173). piRNA dependent 22Gs were also defined as in *Lee et al. (2012)*, as 22G RNAs that mapped in 100 bp windows centered at piRNA target sites, allowing zero or one mismatch.

## H3K9me3 levels in nrde mutants

The following H3K9me3 ChIP seq datasets were used: *nrde-2* (GSM855086), *nrde-3* (GSM932875), *nrde-4* (GSM932876), WT for *nrde-2,–3, −4* (GSM855085), *hrde-1* (GSM1399632), and WT for *hrde-1* (GSM1399631) from [*Gu et al. (2012)*, *Buckley et al. (2012)*, *Ni et al. (2014)*. Datasets were processed as described in the data processing section above. The average signal in each region of

interest was calculated and the H3K9me3 fold change was calculated relative to the matched wild-type dataset. Control repeats (n = 612) have >1.5 fold H3K9me3 levels relative to the genome average, have <1 fold signal for each of the five heterochromatin factors (HPL-2, LIN-13, LIN-61, LET-418 and MET-2) relative to the genome average, and are not upregulated in any of the five mutant strains. A reduction of H3K9me3 at gene and repeat sets of interest was assessed by comparing to all genes or all repeats using a single-sided Mann-Whitney U test.

## Detection of phospho-CHK-1

N2 and *hpl-2* adults grown at 25°C were irradiated at 0, 20, and 100 Gy and recovered for one hour at 25°C. One hour post irradiation, gonads were dissected in 8 µL M9 on slides and freeze-cracked. Gonads were fixed four minutes in 100% methanol followed by twenty minutes in 4% formaldehyde in 1 X PBS. After fixation, gonads were washed two times for ten minutes in 1 X PBS + 0.2% Tween-20 (PBST), blocked for one hour at room temperature in 1% milk in PBST, washed two times for ten minutes in PBST, incubated overnight at 4°C in primary antibody diluted in PBST (1:50 rabbit monoclonal α-phospho-CHK-1, Ser345, 133D, Cell Signalling Technologies, catalogue #2348), washed two times for ten minutes in PBST, incubated 2 hr with secondary antibody (Molecular Probes) and DAPI. Gonads were scored for the presence of phospho-CHK-1 staining using a Zeiss 510 Meta scanning-laser confocal microscope. Counts from individual experiments were pooled to give overall totals, and a two-tailed proportions Z test was used to determine whether there was a difference between N2 and *hpl-2* worms at a specific condition.

## Oocyte chromatin fragmentation assay

L4 N2 and *hpl-2(tm1489)* grown at 20°C were irradiated at 0, 50, and 100 Gy, recovered for 24 hr at 20°C, then fixed in MeOH and DAPI stained. Slides were scored for the number of DAPI bodies in diakinesis oocytes. Oocytes with six DAPI bodies, representing the six bivalent chromosomes, were considered normal; oocytes with other numbers of DAPI bodies, representing chromosome fracturing or clumping, were considered fractured. To determine whether the proportion of oocytes with fractured chromosomes was different between N2 and *hpl-2(tm1489)* worms at a particular condition, a two-tailed proportions Z test was used. Two-tailed P values were calculated using a Z score table.

## Germ line apoptosis measurements

*bcIs39 (Plim7::ced-1::GFP)* expressed in gonadal sheath cells, was used to count engulfed germ line corpses (*Zhou et al., 2001*). Strains containing *bcIs39* in wild type and mutant backgrounds were maintained at 20°C. L4s of each genotype were picked and 48 hr later scored for the number of engulfed apoptotic cells in the gonad. A minimum of 25 gonads per experiment were scored in three independent experiments. The number of apoptotic cells in the germ line observed with the *ced-1*::GFP is higher than the number stained by vital dye or visualized by Nomarski optics because the reporter also marks cells at earlier stages of apoptosis than can be detected by other methods (*Lant and Derry, 2014*; *Lu et al., 2009*). In *lin-13*, *hpl-2*, and *met-2 set-25* strains, full or partial silencing of the GFP transgene reporter occurred in some individuals. These animals were excluded because it was not possible to count cell deaths. Statistical significance was scored using a Mann-Whitney non-parametric test over all the datapoints combined.

## Assessment of abnormal oogenesis

Strains were maintained at 20°C and shifted to 25°C at the L4 stage. Adult germlines were imaged 48 hr later by mounting animals on 3% agarose pads in 5 mM Tetramisole, using a Zeiss widefield upright microscope using Nomarksi optics. Oogenesis was deemed 'abnormal' if oocytes appeared small and rounded, if they were disorganized, or if their cytoplasm had taken on a pronounced curdled texture. Germlines which had mostly or fully disintegrated, and lacked detectable oocytes, were also quantified.

## Sterility interaction tests

Fertility interactions among heterochromatin factors were tested as follows: N2, *let-418(n3536)*, and *lin-13(n770)* were maintained at 20°C. Worms were fed at 20°C from the L4 stage for the following

RNAi clones from *Kamath et al. (2003)*: *hpl-2* (K01G5.2), *met-2* (R05D3.11), and *lin-61* (R06C7.7) or from starved L1s for *lin-13* (sjj2_C03B8.4). RNAi plates were prepared as in *Ahringer, 2006*. Progeny of fed L4s or the fed L1were singled out onto fresh RNAi plates as L4s and total broods assessed by transferring the worms to new plates every day until they stopped laying eggs. Two independent experiments were conducted, with 3–8 total broods counted for each strain/RNAi combination. Three double mutant combinations were also constructed and tested: *lin-61(tm2649); lin-13(n770), lin-61(tm2649); hpl-2(tm1489)*, and *hpl-2(tm1489); let-418(n3536)*. Wild-type N2, single mutants, and double mutant strains were maintained at 20°C and total broods counted.

Fertility interactions between *nrde-2* and *let-418, hpl-2*, or *lin-61* were tested as follows. Wild-type, single mutants, and double mutants were grown at 15°C from starved L1 until the L3 stage, then transferred to 25°C. Total brood size per worm was determined for 12–24 worms per strain across two independent experiments. Genotypes of strains are given in *Supplementary file 2*.

Statistical tests for the above genetic interactions were conducted as follows: Under the null hypothesis that the two genes do not interact to affect fertility, the expected brood size of double mutant (or RNAi knockdown in a single mutant background) is the product of those of the single mutants (or that of the single mutant and the RNAi knockdown in a wild type background) divided by the brood size of the wild-type (or of the wild-type strain grown on control (empty vector) RNAi plates). Similar to (*Baugh et al., 2005*), a t-test was used to test if the observed brood size of double mutants (or RNAi knockdown in a single mutant background) equals the expected brood size under the null hypothesis.

Tests for suppression of sterility of *hpl-2(tm1489), lin-13(n770)*, and *let-418(n3536)* mutants were conducted mutants as follows. The three strains show temperature sensitive sterility. Strains were maintained at 20°C, a temperature at which they are fertile, and starved L1s prepared by bleaching adults to collect embryos and hatching them in M9 buffer for 24 hr at 20°C. Starved L1s were spotted onto RNAi plates prepared as in *Ahringer, 2006*. They were then grown under conditions at which the mutant strain is nearly sterile: *hpl-2* was incubated at 24°C or 25°C, *lin-13* was incubated at 24°C, and *let-418* was incubated for 7.5 hr at 20°C, then shifted to 24°C. After three days, the number of progeny produced by these L1s was counted. RNAi plates were prepared as in *Ahringer, 2006*. The following RNAi clones were from *Kamath et al. (2003)*: *cep-1* (F52B5.5), *spo-11* (T05E11.4), *mirage-A* (K02E7.2 + K02E7.3), *mirage-B* (W03G1.3 + W03G1.4). For each combination of mutant strain and target gene to knockdown, a paired t-test was used to compare the average number of progeny per parent from gene-targeting RNAi plates and matched empty vector plates incubated under the same condition.

Brood sizes of *cep-1(lg12501); hpl-2(tm1489), cep-1(lg12501);let-418(n3536)* and *cep-1(lg12501); lin-13(n770)* double mutants were compared to wild-type and single mutants. Adults raised at 15°C were bleached to obtain embryos and left at 20°C to hatch without food to obtain starved L1s. The starved L1s were fed with OP50 bacteria and immediately shifted to 25°C (for *hpl-2* tests) or fed at 20°C for 7.5 hr before shifting to 25°C (for *let-418* and *lin-13* tests). Experiments were repeated at least twice and total brood sizes were determined for 8 to 40 worms per strain. L1s prepared in the same way were used for growth rate tests, counting the number of adults, L4s, or worms younger than L4 (indicated <L4) after approximately 48 hr post feeding. A single sided t-test was used to test whether the brood size of the *cep-1* double mutant is larger than that of the single heterochromatin mutant.

## piRNA sensor expression

*mjIs144* [*mex-5*p::HIS-58::GFP::piRNA(21UR-1)::*tbb-2*–3'UTR] was used to assess piRNA pathway function (*Ashe et al., 2012*). Synchronized larvae containing *mjIs144* in wild type or mutant backgrounds were maintained at 20°C, and scored for germ line expression 24 hr post-L4 stage. Four experiments were conducted per strain, with a minimum of 25 worms per experiment. GFP was scored using a Zeiss Axioplan two upright widefield microscope, where the level of GFP expression was assessed (none, low, moderate). Moderate expression was scored when GFP was easily detectible in oocytes and pachytene nuclei. Low expression was scored when GFP was just visible in oocyte and pachytene nuclei. In *Figure 6*, silenced represent gonads with no expression and Expressed - GFP(+) represent moderate or low expression. In the case of *met-2*, all expression was in the low category.

## RNA FISH

N2, *hpl-2, lin-13, let-418, and prg-1* young adults were fixed and stained by RNA FISH as described (*Raj et al., 2008*). Stellaris FISH probes targeting MIRAGE1 and *sqv-1* (as an internal control) were obtained from Bioresearch Technologies (Novato, CA). CAL Fluor Red610 was used for MIRAGE1 and Quasar 570 was used for *sqv-1*. From 11–22 individuals per strain were scored.

## Datasets

Datasets generated in this paper are available at GEO accession GSE87524.

## Acknowledgements

We are grateful to A Akay for helpful comments on the manuscript. This work was supported by Wellcome Trust Senior Research Fellowships to JA (054523 and 101863). AM was supported by a fellowship from the Canadian Institutes of Health Research, DA by a Darwin Trust scholarship, and AS by an HFSP postdoctoral fellowship. EM was supported by grants from the Wellcome Trust (104640/Z/14/Z) and Cancer Research UK (C13474/A18583). JA and EM also acknowledge support by core funding from the Wellcome Trust (092096) and Cancer Research UK (C6946/A14492).

## Additional information

### Competing interests

JA: Reviewing editor, *eLife*. The other authors declare that no competing interests exist.

### Funding

| Funder | Grant reference number | Author |
| --- | --- | --- |
| Wellcome | 054523 | Julie Ahringer |
| Wellcome | 101863 | Julie Ahringer |
| Canadian Institutes of Health Research | | Alicia N McMurchy |
| Cancer Research UK | C13474/A18583 | Eric A Miska |
| Human Frontier Science Program | | Alexandra Sapetschnig |
| Wellcome | 104640/Z/14/Z | Eric A Miska |

The funders had no role in study design, data collection and interpretation, or the decision to submit the work for publication.

### Author contributions

ANM, PS, TG, Conceptualization, Formal analysis, Investigation, Visualization, Methodology, Writing—original draft, Writing—review and editing; BW, YD, DA, AA, PK-Z, AS, Formal analysis, Investigation; NH, Formal analysis, Investigation, Visualization, Methodology; EAM, Supervision, Funding acquisition; JA, Conceptualization, Formal analysis, Supervision, Funding acquisition, Investigation, Visualization, Writing—original draft, Writing—review and editing

### Author ORCIDs

Alicia N McMurchy, http://orcid.org/0000-0002-7033-8790
Przemyslaw Stempor, http://orcid.org/0000-0002-9464-7475
Ni Huang, http://orcid.org/0000-0001-8849-038X
Eric A Miska, http://orcid.org/0000-0002-4450-576X
Julie Ahringer, http://orcid.org/0000-0002-7074-4051

## Additional files

### Supplementary files

• Supplementary file 1. LET-418, LIN-13, HPL-2, LIN-61, and MET-2 are required for normal fertility.

• Supplementary file 2. Strains used in this study.

• Supplementary file 3. Antibodies used in this study.

### Major datasets

The following dataset was generated:

| Author(s) | Year | Dataset title | Dataset URL | Database, license, and accessibility information |
|---|---|---|---|---|
| McMurchy AM, Stempor P, Gaarenstroom T, Wysolmerski B, Dong Y, Aussianikava D, Appert A, Huang N, Kolasinska-Zwierz K, Sapetschnig A, Miska E, Ahringer J | 2017 | A team of heterochromatin factors collaborates with small RNA pathways to combat repetitive elements and germline stress | https://www.ncbi.nlm.nih.gov/geo/query/acc.cgi?acc=GSE87524 | Publicly available at the NCBI Gene Expression Omnibus (accession no: GSE87524) |

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
