## [Decision Letter]

Thank you for submitting your article "A team of heterochromatin factors collaborates with small RNA pathways to combat repetitive elements and germline stress" for consideration by *eLife*. Your article has been favorably evaluated by Jessica Tyler (Senior Editor) and three reviewers, one of whom, Edith Heard (Reviewer #1), is a member of our Board of Reviewing Editors.

The reviewers have discussed the reviews with one another and the Reviewing Editor has drafted this decision to help you prepare a revised submission.

Summary:

In this paper, McMurchy, Stempor et al., investigate the relationship between repetitive elements and heterochromatin and the interaction between heterochromatin factors and small RNA pathways in *C. elegans*. They systematically investigate a series of five heterochromatin factors (HPL-2/HP1, LIN-13, MBT-repeat protein LIN-61, LET-418/Mi-2, and the H3K9me2 histone methyltransferase MET-2/SETDB1) that have fertility phenotypes when mutated, for their binding profiles and for their impact on repeat elements, DNA repair and apoptosis in the germ line. Their results show that these heterochromatin proteins repress the expression (and potentially the noncoding activity – recombination) of repetitive DNA elements in *C. elegans* germ cells. They also investigate the relationships between these heterochromatin factors and the piRNA and nuclear RNAi pathways. Their study reveals that heterochromatin proteins and small RNA pathways display overlapping as well as unique control of repetitive elements in the germ line.

This work brings several important new insights:

It provides solid genetic and functional links between key factors previously shown to be associated with heterochromatin and known to lead to sterility when mutated.

It demonstrates that derepression of repetitive elements in heterochromatin mutants is one of the reasons these mutants are sterile.

It describes the genomic co-localisation of these five heterochromatin factors, particularly at repetitive elements associated with H3K9 methylation.

It provides new insights into the role that binding of these heterochromatin factors has at repeats, as RNA-seq revealed that although most repeats are derepressed to a small extent, the most affected repetitive elements in heterochromatin mutants were those encoding for transposases – these being the most critical for protection of the germ line from repeat reactivation.

It provides a convincing demonstration that repeat element upregulation is a contributing factor to infertility in heterochromatin mutants by using RNAi knockdown of the repetitive element MIRAGE1 which rescues sterility.

It demonstrates that the piRNA pathway is impeded upon loss of heterochromatin factors by demonstrating the derepression of a piRNA sensor in heterochromatin mutants.

It also shows that the nuclear RNAi pathway is largely independent of the heterochromatin and piRNA pathways, as a very different spectrum of repetitive element derepression was found in *nrde-2* mutants compared to the *prg-1* piRNA mutant or heterochromatin mutants.

It allows the conclusion, from genetic tests, that these alternate pathways are acting redundantly at some repeats, and uniquely at others.

Overall this is an elegant study, combining genetic and genomic approaches to investigate the role that heterochromatin factors play in repetitive element germ line control and fertility in *C. elegans*. It provides novel mechanistic insights into the multiple levels at which heterochromatin factors can act, not only in silencing repeats but also participating in DNA repair pathways to maintain genome integrity. The experiments are extremely well conducted, clearly presented and discussed. The interactions between heterochromatin factor pathways and small RNA pathways are still rather tenuous and could be further explored as suggested below.

Essential points:

1) Does upregulation of repetitive elements through loss of heterochromatin disable or overload the piRNA pathway? The authors could address this by small RNA-seq in the different heterochromatin mutant backgrounds, and would expect to see a change in the 21U/22G population if the piRNA pathway is overloaded (i.e. upregulation of a repetitive element leads to a bias in the 22G population in the mutant and a decrease in other 22Gs), or a loss of 21Us if piRNAs are completely eliminated. A simpler experiment might be to see whether any of the RNAi treatments in Figure 4 restore piRNA sensor silencing in the heterochromatin factor mutant backgrounds to link repetitive element expression and/or DNA repair to piRNA pathway function.

2) The authors could use ChIP-seq or immunofluorescence for histone modifications in heterochromatin factor mutant backgrounds in order to demonstrate the loss of heterochromatin marks at repetitive elements. Without such experiments, the connection between heterochromatin proteins and small RNA pathways remains tenuous. If the authors cannot add such experiments they must at least explain and discuss this.

3) In Figure 4: Is the increase in fertility in response to *mirage, cep-1*, and *spo-11* knockdown (KD) a general phenomenon or specific to heterochromatin mutants? The authors could compare the brood size of wt cells with and without KD of the above genes. Measure fold change in wt plus or minus KD and compare to the fold change seen in mutants.

4) Figure 5: how specific is the *cep-1*-mediated promotion of apoptosis in response to DNA damage? Do cells with no DDR also show an increase in survival if *cep-1* is knocked down? The authors could compare the survival of wt cells with and without *cep-1* knockdown and compare this to the fold change observed in mutants.

5) The authors should highlight the novelty of their own findings in this manuscript by discussing their contributions in the context of what has been previously published in nuclear RNAi, piRNA and heterochromatin in *C. elegans* and other organisms e.g. Ashe et al. 2012; Gu et al. 2012; Burton et al. 2011; Buckley et al. 2012; Ni et al. 2014).

---

## [Author Response]

Essential points:

1) Does upregulation of repetitive elements through loss of heterochromatin disable or overload the piRNA pathway? The authors could address this by small RNA-seq in the different heterochromatin mutant backgrounds, and would expect to see a change in the 21U/22G population if the piRNA pathway is overloaded (i.e. upregulation of a repetitive element leads to a bias in the 22G population in the mutant and a decrease in other 22Gs), or a loss of 21Us if piRNAs are completely eliminated. A simpler experiment might be to see whether any of the RNAi treatments in Figure 4 restore piRNA sensor silencing in the heterochromatin factor mutant backgrounds to link repetitive element expression and/or DNA repair to piRNA pathway function.

Small RNA sequencing was previously done for *hpl-2* mutants in Ashe et al., 2012, where they found normal levels of piRNAs targeted to the piRNA sensor and to a few endogenous target genes. We have extended this analysis to the whole genome and find that piRNA levels are normal in *hpl-2* mutants. We also tested whether piRNA dependent 22G RNAs were produced in *hpl-2* mutants using the method in Lee et al. 2012 to define piRNA targets. We found that piRNA dependent 22G RNA levels were reduced in *prg-1* mutants, as expected, but levels were normal in *hpl-2* mutants. These results suggest that *hpl-2* acts downstream of piRNA biogenesis and subsequent 22G production. We have added these data to the paper in Figure 6—figure supplement 1.

As suggested, we also tested whether RNAi of *cep-1, spo-11,* or MIRAGE1 elements restored piRNA sensor silencing in heterochromatin mutants and found that none did, arguing that the role of heterochromatin factors in the piRNA pathway is independent of these genes (see Figure 9). Because this experiment does not conclusively show that the affected pathways are not involved, we have decided not to include this figure in the paper.

Author response image 1.Quantification of piRNA sensor expression in wild type and heterochromatin mutants *hpl-2, let-418* and *lin-61.*Animals were fed on indicated RNAi bacteria from L1 at 20°C and scored as 1 day old adults, similar to Figure 6. A minimum of 15 worms were scored over 2 independent experiments. None of the RNAi treatments resilenced the sensor.**DOI:**
http://dx.doi.org/10.7554/eLife.21666.036

2) The authors could use ChIP-seq or immunofluorescence for histone modifications in heterochromatin factor mutant backgrounds in order to demonstrate the loss of heterochromatin marks at repetitive elements. Without such experiments, the connection between heterochromatin proteins and small RNA pathways remains tenuous. If the authors cannot add such experiments they must at least explain and discuss this.

We agree that better understanding the connection between heterochromatin proteins and small RNA pathways is important. The suggested experiments would address the role of heterochromatin proteins in regulating histone modifications but not the link to small RNA pathways. However, were able to make use of published H3K9me3 ChIP seq data in four nuclear RNAi mutant backgrounds to explore this connection. Analysing these datasets, we found that germ line nrde pathway mutants (*hrde-1, nrde-2, nrde-4*) had decreased H3K9me3 marking at heterochromatin regulated loci, but there was no change in *nrde-3* mutants, which are defective in nuclear RNAi in the soma (Figure 7—figure supplement 2 and Figure 7—figure supplement 3). The decrease in H3K9me3 levels on heterochromatin targets that are not upregulated in *nrde-2* mutants, supports redundancy in target regulation.

3) In Figure 4: Is the increase in fertility in response to mirage, cep-1, and spo-11 knockdown (KD) a general phenomenon or specific to heterochromatin mutants? The authors could compare the brood size of wt cells with and without KD of the above genes. Measure fold change in wt plus or minus KD and compare to the fold change seen in mutants.

Because mutation of *cep-1* or *spo-11* does not cause an increased brood size (Rinaldo et al. 2002 and Figure 4), the effect of RNAi knockdown on heterochromatin mutants is not due to a general increase in fertility. We have strengthened the fertility suppression results by testing whether mutation of *cep-1*, like RNAi, suppressed the decreased fertility of heterochromatin mutants. We found that *hpl-2; cep-1, lin-13; cep-1*, and *let-418; cep-1* double mutants all had larger brood sizes than the corresponding heterochromatin single mutants, confirming the RNAi results. In addition, we observed that mutation of *cep-1* also partially rescued the somatic growth defect of the mutants, suggesting that the interaction is not limited to prevention of cell death. Because p53 is important for mediating DNA damage signaling, the results suggest that such signaling may underlie heterochromatin factor defects. We include the new *cep-1* double mutant results in Figure 4.

4) Figure 5: how specific is the cep-1-mediated promotion of apoptosis in response to DNA damage? Do cells with no DDR also show an increase in survival if cep-1 is knocked down? The authors could compare the survival of wt cells with and without cep-1 knockdown and compare this to the fold change observed in mutants.

See response to point 3 above. In addition to promoting apoptosis in the germ line in response to DNA damage, *cep-1*/p53 is also responsible for the slow growth caused by loss of DNA damage checkpoint protein CLK-2/TEL2 (Derry et al., 2007). Similarly, we observed that mutation of *cep-1* suppressed the slow growth phenotypes of *hpl-2, lin-13*, and *let-418* (Figure 4). We do not know if fertility suppression by loss of *cep-1* is due to prevention of apoptosis or another stress response. To reflect this, we now write: “The increase in fertility upon *cep-1*/p53 inhibition may be a direct consequence of reduced germ line apoptosis, or alternatively the effect may be indirect, by preventing DNA damage signalling or improving growth rate.”

5) The authors should highlight the novelty of their own findings in this manuscript by discussing their contributions in the context of what has been previously published in nuclear RNAi, piRNA and heterochromatin in C. elegans and other organisms e.g. Ashe et al. 2012; Gu et al. 2012; Burton et al. 2011; Buckley et al. 2012; Ni et al. 2014).

We have rewritten and expanded the Introduction and Discussion to better explain previous studies and the advances made by our study.